# Optimal Causal Representations and the Causal Information Bottleneck

## Abstract

To effectively study complex causal systems, it is often useful to construct representations that simplify parts of the system by discarding irrelevant details while preserving key features. The Information Bottleneck (IB) method is a widely used approach in representation learning that compresses random variables while retaining information about a target variable. Traditional methods like IB are purely statistical and ignore underlying causal structures, making them ill-suited for causal tasks. We propose the Causal Information Bottleneck (CIB), a causal extension of the IB, which compresses a set of chosen variables while maintaining causal control over a target variable. This method produces representations which are causally interpretable, and which can be used when reasoning about interventions. We present experimental results demonstrating that the learned representations accurately capture causality as intended.

## 1 Introduction

Natural systems typically consist of a vast number of components and interactions, making them complex and challenging to study. When investigating a specific scientific question, which is often of a causal nature, it is frequently possible to disregard many of these details, as they have a negligible impact on the outcome. These details can then be abstracted away. A classic example (Rubenstein et al., 2017; Chalupka et al., 2017) is the relationship between particle velocities, temperature, and pressure. To control the pressure on the walls of a room, it would be necessary to influence the velocities of the approximately $10^{23}$ particles in the room. However, controlling the velocity of each individual particle is not required to achieve this. Instead, manipulating the high-level variable of temperature is sufficient. For an ideal gas, temperature is directly proportional to the average kinetic energy of the particles in the gas (Blundell & Blundell, 2010), and modifying the temperature alters the particle velocities accordingly.

Methods that disregard the causal structure of a system when constructing abstractions may yield results that are uninformative or even misleading, particularly when the objective is to manipulate the system or gain causal insights. The following running example will serve as a useful illustration of the potential drawbacks of neglecting causal considerations when learning representations. Consider a mouse gene with four positions $s_1, s_2, s_3, s_4$ under study[1] where nucleotides may be mutated, corresponding to the binary variables $X_i, \ i = 1, \ldots, 4$, which indicate whether there is a mutation at position $s_i$. These mutations can interact in a complex manner with respect to a phenotype of interest $Y$, say the body mass of the mouse. This type of complex interaction is known as epistasis (Phillips, 2008). One could create a abstraction $T$ of $X_1, X_2, X_3, X_4$ that would be blind to differences between mutation configurations $(X_1, X_2, X_3, X_4)$ that do not provide information about $Y$. With the current description, it may seem that there is no need for considering causality, and that one could simply use a purely statistical method to learn a good "epistasis gauger" $T$. This is, however, not the case, since the $X_i$ and $T$ are typically confounded by the population structure, that is, "any form of relatedness in the sample, including ancestry differences or cryptic relatedness" (Sul et al., 2018). Based on the discussion in Sul et al. (2018), we consider the population structure variable $S$ encoding the strain of the mice (laboratory vs wild-derived strains), which is associated with both distinct $X$ and body masses $Y$ (high in laboratory strains and low in wild-derived strains).

---

[1]The number of locations in the genome that may affect $Y$ may be orders of magnitude larger than what we consider here. We use small numbers to illustrate the problem.

A possible resulting causal graph can be seen in Figure 3. As a result, mutations that are more prevalent in mice from the wild-derived strain will exhibit a strong correlation with low body mass, even if there is no underlying causal relationship between them. This is an example of how population structure can work as a confounder. In humans, population structure in the form of ancestry plays a similar role to the mouse strains here.

The construction of abstractions, in the form of representations, has been a prominent area of research in *non-causal* machine learning for many years, with both theoretical and applied contributions. A "good" representation should facilitate the learning task (Goodfellow et al., 2016), achieving multiple benefits: it should conserve space, simplify computations, and accelerate processing, while also uncovering the essential aspects of $X$, which may help with interpretability. It may also make the learning task easier because highly flexible machine learning algorithms often struggle with overfitting, which is exacerbated by the presence of unimportant information (Bishop, 2006). It is essential to recognize that the determination of which information is spurious critically depends on the specific task at hand. Building a representation is therefore a balancing act between keeping enough information for the task and avoiding including unnecessary details. This is the insight which Tishby et al. (2000) build on. The authors formalize the learning of the optimal representation $T$ for an input variable $X$ and a chosen target variable $Y$ as a minimization problem. This problem involves finding a balance between compressing $X$ as much as possible and maintaining as much information about $Y$ as possible. The functional proposed by Tishby et al. (2000) to be minimized is called the *Information Bottleneck (IB) Lagrangian*, and it has inspired a significant body of work in representation learning (see for example Alemi et al. (2016); Kolchinsky et al. (2018; 2019); Tishby & Zaslavsky (2015); Achille & Soatto (2018)).

The IB method is not aware of causality. The learned representations cannot be used to reason causally about the system. In particular, when $X$ and $Y$ are heavily confounded, the IB method will create a representation $T$ that preserves the information $X$ has about $Y$, but a significant portion of this information is spurious from a causal perspective. This not only leads to sub-optimal compression but can also result in misleading conclusions, where the values of $T$ are mistakenly interpreted as corresponding to meaningful interventions. The importance of constructing appropriate causality-aware representations and abstractions has been recognized previously (Schölkopf et al., 2021; Beckers & Halpern, 2019; Rubenstein et al., 2017; Chalupka et al., 2017; Höltgen, 2021; Jammalamadaka et al., 2023), but the successful construction of such representations remains an open challenge (see Section 9).

In this paper, we present a new method for finding representations of a set of input variables $X$ which retain the causal information between $X$ and a specified target variable $Y$. Namely, we introduce a causal version of the IB Lagrangian designed to attain its minima when the representation $T$ of $X$ compresses $X$ as much as possible while maintaining as much *causal* control over $Y$ as desired, with the trade-off between these properties governed by a parameter $\beta$. We derive the *Causal Information Bottleneck (CIB) Lagrangian* by first establishing an axiomatic description of what constitutes an *optimal causal representation*. Similar to the original Information Bottleneck paper (Tishby et al., 2000), we take all variables to be discrete, and we assume that we have access to the joint distribution over all variables. Additionally, the causal graph containing $X$ and $Y$ is taken to be known.

The contributions of this paper can be stated as follows:

- We propose an axiomatic definition of optimal causal representation, which naturally extends previous definitions of optimal representation.

- We derive a causal version of the Information Bottleneck Lagrangian from those axioms, thereby formulating the problem of optimal causal representation learning as a minimization problem. To the best of our knowledge, this is the first time that representations of variables are designed to be interventionally relevant for the target variable.

- We propose a definition of equivalence for representations and an information-theoretical metric to capture it in the deterministic case.

- We derive a backdoor criterion formula for representations, enabling us to compute a representation's post-intervention distribution from observational data.

This paper is structured as follows: In Section 2 we briefly review the necessary concepts from the theories of causal models, causal information, and the information bottleneck method. Section 3

defines optimal causal representations and formulates the constraint minimization problem whose solutions are the optimal causal representations. The causal information bottleneck Lagrangian is introduced in Section 4, which is used to solve the aforementioned minimization problem. Section 5 defines interventions on representations, and Section 6 builds on this to derive a backdoor adjustment formula for representations. In Section 7 we define when two representations are equivalent, and propose an informational-theoretical metric that can detect it in the deterministic case. These elements are then used when learning the optimal causal representations in 3 different experiments, whose results are reported in Section 8. Finally, Sections 9 and 10 discuss related work and the obtained results, and propose future research directions. All proofs can be found in the Appendix, which also contains various supplements to the main text. The code repository containing the experiments can be found in the file submitted alongside the paper.

## 2 Preliminaries

### 2.1 Causal Models and Interventions

A Structural Causal Model (SCM) provides a representation of a system's causal structure, analogous to a Bayesian network but with a causal interpretation. An SCM $\mathfrak{C} = (\mathbf{V}, \mathbf{N}, S, p_{\mathbf{N}})$ is comprised of endogenous variables $\mathbf{V}$, exogenous variables $\mathbf{N}$, deterministic functions between them $S$, and a distribution over the noise variables $p_{\mathbf{N}}$. Each SCM $\mathfrak{C}$ has an underlying DAG $G^{\mathfrak{C}}$, called its causal graph. Each node in the DAG corresponds to an endogenous variable, while the edges stand for the causal relationships between them. We denote the parents and children of an endogenous variable $X$ by $\mathrm{Pa}(X)$ and $\mathrm{Ch}(X)$, respectively. Furthermore, we denote the range of a random variable $X$ by $R_X$ and its support by $\mathrm{supp}(X)$. The value of each endogenous variable is determined by a deterministic function of its parent variables and an independent exogenous variable, which accounts for the system's randomness. A key feature of SCMs is their capacity to model interventions on a variable $X$, which involve altering the variable's generating process. This results in a new SCM with its own distribution, reflecting the system's state post-intervention. The most common type of intervention is an atomic intervention, where a variable $X$ is set to a specific value $x$, effectively severing its connection to its parents and assigning a fixed value instead. We denote such an intervention by $do(X = x)$, the resulting SCM by $\mathfrak{C}^{do(X=x)}$, and the post-intervention joint distribution of a set of variables $W$ by $p_W^{do(X=x)}(w)$ or $p(w \mid do(X = x))$. For more details, see Appendix A.3.

### 2.2 Causal Entropy and Causal Information Gain

We now introduce two concepts: causal entropy and causal information gain, both of which are fundamental to our method. For more details, see Appendix A.2, or refer to the work by Simoes et al. (2023). The causal entropy $H_c(Y \mid do(X))$ measures the average uncertainty remaining about the variable $Y$ after we intervene on the variable $X$. This concept is closely related to conditional entropy but adapted for situations where interventions on $X$, as opposed to conditioning on $X$, are considered. The causal information gain $I_c(Y \mid do(X))$ extends the idea of mutual information to the causal domain. It quantifies the reduction in uncertainty about $Y$ after intervening on $X$, offering a measure of the causal control that $X$ exerts over $Y$. Essentially, it tells us how much more we know about $Y$ due to these interventions on $X$.

### 2.3 The Information Bottleneck Lagrangian

Let $X$ be a random variable. By a representation $T$ of $X$, we mean a variable that can only depend on $X$, whether deterministically or stochastically. This means in particular that $T$ must be independent of $Y$ when conditioning on $X$. This generalizes the notion of representation used by Achille & Soatto (2018) for cases with more variables than only $X$ and $Y$. Furthermore, a representation $T$ is characterized by its encoder, which codifies how $T$ depends on $X$. We formalize this as follows:

**Definition 1** (Representation). *A random variable $T$ is a* representation *of a random variable $X$ if $T$ is a function of $X$ and an independent noise variable. The* encoder *of the representation $T$ is the function $q_{T|X} \colon R_T \times R_X \to [0, 1]$ such that $q_{T|X}(t \mid x)$ is the conditional probability $q(t \mid x)$ of $T = t$ given $X = x$.*

In Tishby et al. (2000), the authors aim at finding a representation $T$ of $X$ which "keeps a fixed amount of meaningful information about the relevant signal $Y$ while minimizing the number of bits from the original signal $X$ (maximizing the compression)." This is accomplished by introducing the Information Bottleneck Lagrangian $\mathcal{L}_{\text{IB}}^{\beta}[q_{T|X}] = I(X;T) - \beta I(Y;T)$, where $\beta$ is a non-negative parameter which manages the trade-off between compression (as measured by $I(X;T)$) and sufficiency (as measured by $I(Y;T)$). The problem of finding such a representation becomes then the problem of finding an encoder $q_{T|X}$ which minimizes the $\mathcal{L}_{\text{IB}}^{\beta}$. They minimize this Lagrangian by adapting the Blahut-Arimoto algorithm from rate distortion theory (Blahut, 1972) to their case, resulting in a coordinate-descent optimization algorithm. Crucially, this adaptation relies on the fact that, similarly to Blahut (1972), searching for the zero-gradient points of the Lagrangian results in self-consistency equations that can be exploited to construct a coordinate-descent algorithm. This construction hinges on viewing mutual information as a KL divergence.

The IB method proposed by Tishby et al. (2000) effectively solves the following minimization problem:

$$\underset{q_{T|X} \in \mathbb{R}^{|R_T| \cdot |R_X|}}{\arg\min} I(X;T) \qquad \text{s.t.} \qquad \begin{cases} \forall x \in R_X, q_{T|X=x} \in \Delta^{|R_T|-1} \\ I(Y;T) = D \end{cases}, \qquad (1)$$

where $\Delta^{|R_T|-1}$ is the probability simplex, and $D$ belongs to the sufficiency values compatible with the chosen $\beta$. Notice that in particular $q_{T|X}$ is then constrained to $\Delta \coloneqq \bigtimes_{x \in R_X} \Delta^{|R_T|-1}$ (see Appendix E for details).

## 3    OPTIMAL CAUSAL REPRESENTATIONS

For the remainder of the paper, let $X \subseteq \mathbf{V}$ be a set of endogenous variables of an SCM $\mathfrak{C} = (\mathbf{V}, \mathbf{N}, S, p_{\mathbf{N}})$, $T$ be a representation of $X$ with encoder $q_{T|X}$, and $t$ be an element of $R_T$.

In a natural extension of the qualitative description of optimal representation in Section 2.3 to the causal context, our problem can be described as finding representations $T$ of $X$ which retain a chosen amount $D$ of *causal* information about the relevant signal $Y$ while minimizing the information that $T$ preserves about $X$. We propose an axiomatic characterization of optimal causal representation to formally capture this description using information-theoretical quantities. This can also be seen as a causal variant of the characterization of optimal representation from Achille & Soatto (2018). Since we use $I_c(Y \mid do(T))$ to measure the causal information that $T$ has about $Y$, and $I(X;T)$ is the information that $T$ keeps about $X$, the result is the following:

**Definition 2** (Optimal Causal Representation). *A optimal causal representation of $X$ at sufficiency $D$ is a representation $T$ of $X$ such that:*

   *(C1)  $T$ is interventionally $D$-sufficient for the task $Y$, i.e., $I_c(Y \mid do(T)) = D$.*

   *(C2)  $I(X;T)$ is minimal among the variables $T$ satisfying (C1).*

We can then formulate the problem of finding an optimal causal representation as the following minimization problem:

$$\underset{q_{T|X} \in \mathbb{R}^{|R_T| \cdot |R_X|}}{\arg\min} I(X;T) \qquad \text{s.t.} \qquad \begin{cases} \forall x \in R_X, q_{T|X=x} \in \Delta^{|R_T|-1} \\ I_c(Y \mid do(T)) = D \end{cases}. \qquad (2)$$

Recall that the IB Lagrangian can be seen as arising from the minimization problem in Equation (1). Likewise, the CIB Lagrangian will be introduced to solve the minimization problem in Equation (2).

## 4    THE CAUSAL INFORMATION BOTTLENECK

We can find the solution(s) to Equation (2) within the minimizers of a Lagrangian. Specifically, we can minimize (for some chosen $X$, $Y$ and $R_T$) the *causal information bottleneck Lagrangian* $\mathcal{L}_{\text{CIB}}^{\beta}$ defined as follows:

**Definition 3** (Causal Information Bottleneck). *The* causal information bottleneck (CIB) Lagrangian *with trade-off parameter $\beta \geq 0$ is the function $\mathcal{L}_{\text{CIB}}^{\beta} : \mathbb{R}^{|R_T|} \times \mathbb{R}^{|R_X|} \to \mathbb{R}$ given by*

$$\mathcal{L}_{\text{CIB}}^{\beta}[q_{T|X}] \coloneqq I(X;T) - \beta I_c(Y \mid do(T)). \qquad (3)$$

The trade-off parameter $\beta$ is the Lagrange multiplier for the interventional sufficiency constraint, and is taken to be fixed. Notice that the constraint $q_{T|X} \in \Delta$ still needs to be enforced explicitly. As for the IB Lagrangian, fixing $\beta$ restricts the values $D$ that $I_c(Y \mid do(T))$ can take when minimizing $\mathcal{L}_{\text{CIB}}^{\beta}$ (Kolchinsky et al., 2018). In general, we expect larger $\beta$ to favor interventional sufficiency over compression, resulting in smaller values of $D$ and larger values of $I(X;T)$. We assume $\beta$ to be a non-negative real number in view of its interpretation as a trade-off parameter.

As discussed in Appendix E, developing a method to minimize Equation (3) analogous to that of Tishby et al. (2000) is challenging, if not infeasible. Instead, we employ constrained optimization iterative local search algorithms based on gradient descent to find the minima of $\mathcal{L}_{\text{CIB}}^{\beta}$ while adhering to the probability simplices constraint $q_{T|X} \in \Delta$.

## 5 INTERVENTIONS ON REPRESENTATIONS

As discussed in Section 1, there is often interest in creating representations that can be intervened on. An intervention on a representation $T$ of $X$ must correspond to interventions on the low-level variables $X$. Therefore, an intervention on $T$ will induce a distribution over the possible interventions on $X$, which we denote by $p^\star(x \mid t)$ and will refer to as the "intervention decoder". We require that $p^\star(x \mid t)$ be compatible with the encoder $q(t \mid x)$, in the sense that both must agree with a common joint distribution over $T$ and $X$.

**Definition 4** (Intervention Decoder). *The* intervention decoder $p^\star(x \mid t)$ *for the representation $T$ of $X$ is computed from the encoder $q_{T|X}$ using the Bayes rule with a chosen prior $p^\star(x)$, that is,*

$$p^\star(x \mid t) := \frac{q(t \mid x)p^\star(x)}{\sum_{\dot{x}} q(t \mid \dot{x})p^\star(\dot{x})} \tag{4}$$

Notice that a choice of prior over the possible atomic distributions on $X$ still needs to be made. In practice, we will make the choice that $p^\star(x)$ be uniform, so that $p^\star(x \mid t) = \frac{q(t|x)}{\sum_{\dot{x}} q(t|\dot{x})}$.

In order to compute the effect of intervening on $T$ on the SCM variables $\mathbf{V}$, we compute the effect of the atomic interventions $do(X = x)$, and weight them with the likelihood of that intervention using the intervention decoder.

**Definition 5** (Representation Intervention). *The* representation intervention distribution $p_{\mathbf{V}}^{do(T=t)}$ *is the weighted average of the atomic intervention distributions $p_{\mathbf{V}}^{do(X=x)}$ over $x \in R_X$, where the weights are given by the intervention decoder $p^\star(x \mid t)$. That is,*

$$p_{\mathbf{V}}^{do(T=t)}(\mathbf{v}) = p(\mathbf{v} \mid do(T = t)) := \sum_x p^\star(x \mid t)p_{\mathbf{V}}^{do(X=x)}(\mathbf{v}). \tag{5}$$

## 6 BACKDOOR ADJUSTMENT FOR REPRESENTATIONS

In order to learn the optimal causal representation, the learning algorithm will need to estimate $\mathcal{L}_{\text{CIB}}^{\beta} = I(X;T) - \beta I_c(Y \mid do(T))$ at each iteration step, making use of the encoder $q_{T|X}$ at that iteration and the joint $p_{\mathbf{V}}$. The compression term $I(X;T)$ can be directly computed using $p_X$ and the encoder, while the interventional sufficiency term $I_c(Y \mid do(T))$ demands the computation of $p(y \mid do(T = t))$. Equation (15) allows us to write $p(y \mid do(T = t))$ in terms of the encoder $q(t \mid x)$ and the intervention distributions $p(y \mid do(X = x))$. Hence, $p(y \mid do(T = t))$ is identifiable (*i.e.*, computable from the joint $p_{\mathbf{V}}$) if $p(y \mid do(X = x))$ also is. Satisfaction of the *backdoor criterion* is one of the most common ways to have identifiability. In case a set $Z \subseteq \mathbf{V}$ exists satisfying the backdoor criterion relative to $(X, Y)$, then $p(y \mid do(X = x))$ is identifiable, and is given by the backdoor adjustment formula (Pearl, 2009). From the definition of representation intervention and the backdoor adjustment formula, one can derive a backdoor adjustment formula for representations.

**Proposition 6** (Backdoor Adjustment Formula for Representations). *Let $T$ be a representation of $X$ with encoder $q(t \mid x)$, and $Z$ be a set of variables of $\mathfrak{C}$ satisfying the backdoor criterion relative to $(X, Y)$ in $\mathfrak{C}$. Then the representation intervention distribution for $Y$ is identifiable and is given*

*by:*

$$p(y \mid do(T = t)) = \sum_z p(z) \sum_x p(y \mid x, z) p^\star(x \mid t)$$

$$= \sum_z p(z) \sum_x p(y \mid x, z) \frac{q(t \mid x) p^\star(x)}{\sum_{\dot{x}} q(t \mid \dot{x}) p^\star(\dot{x})}, \tag{6}$$

*where $p^\star(x \mid t)$ is the intervention decoder for $T$, and $p^\star(x)$ is the prior over interventions on $X$.*

*Remark* 7 (Interpretation of the backdoor adjustment formula for representations). Since $Z$ meets the backdoor criterion for $(X, Y)$, the probability $p(y \mid x, z)$ can be seen as the probability of observing $Y = y$ in the subpopulation $Z = z$ given that one does $do(X = x)$. Furthermore, $p^\star(x \mid t)$ is the probability that $X$ is set to $x$, given that $T$ is set to $t$. Thus, the sum over $x$ can be seen as the average effect of $X$ on $Y$ in the subpopulation $Z = z$ and given that $T$ is set to $t$. Hence the post-intervention probability $q(y \mid do(t))$ is the average over all subpopulations $Z = z$ of the average effects of $X$ on $Y$, given that $T = t$.

Proposition 6 allows us to express the interventional sufficiency term of the CIB in terms of the encoder and the joint $p_{XYZ}$ as long as there is a set $Z$ which satisfies the backdoor criterion relative to $(X, Y)$. The expression for $I_c(Y \mid do(T))$, and thus for the CIB, will have the same form in all such cases. From now on, we will assume that such a set $Z$ exists. Note that this imposes a restriction solely on the graph structure.

**Assumption 8.** *A set of random variables $Z$ satisfying the backdoor criterion (Pearl, 2009) relative to $(X, Y)$ in the SCM $\mathfrak{C}$ can be conditioned on.*

Assumption 8 is introduced for the sake of convenience and simplifying the algorithm. If violated, $p(y \mid do(t))$ may still be identifiable. In that case, one can make use of do-calculus (Peters et al., 2017) to obtain an expression for $p(y \mid do(x))$, and thus also for $p(y \mid do(t))$, which will hold for that specific causal graph.

## 7 COMPARING REPRESENTATIONS

After learning a representation $T_1$, we may want to compare it with another representation $T_2$, which could either be one learned earlier or one considered as the ground truth. Simple equality of their encoders is not an appropriate criterion for this comparison. Two representations might have different encoders and still be "equivalent" in the sense that they coincide when the values of the representations are relabeled. This is especially apparent in the case of completely deterministic representations. For example, consider two binary representations $T_1$ and $T_2$ of a low-level variable $X$ with range $R_X = \{0, 1, 2\}$, defined by deterministic functions $\phi_{T_1}$ and $\phi_{T_2}$. Suppose $\phi_{T_1}$ maps 0 and 1 to 0, and 2 to 1, while $\phi_{T_2}$ maps 0 and 1 to 1, and 2 to 0. Intuitively, $T_1$ and $T_2$ represent the same representation because relabeling the values of $T_1$ (*i.e.*, swapping 0 and 1) yields a representation that produces the same values as $T_2$ for the same low-level values of $X$. Formally, and extending to the non-deterministic case, this is to say that equivalence arises whenever the conditional distributions are identical up to a bijection $\sigma$ of the values of the $T_i$. This leads to the following definition:

**Definition 9** (Equivalent representations). *Two representations $T_1$ and $T_2$ of $X$ are* equivalent *if there is a bijection $\sigma\colon \mathrm{supp}(T_1) \to \mathrm{supp}(T_2)$ such that $\forall t_1 \in \mathrm{supp}(T_1), x \in \mathrm{supp}(X)$, $q_{T_1|X}(t_1 \mid x) = q_{T_2|X}(\sigma(t_1) \mid x)$, where $q_{T_1|X}, q_{T_2|X}$ are the encoders for $T_1$ and $T_2$. We then write $T_1 \cong T_2$.*

One can show that $\cong$ is an equivalence relation (see Proposition 19). We call *abstraction* an equivalence class of $\cong$, that is, the elements of $\Delta /\!\cong$. In practice, it is unlikely that two representations will be exactly equivalent. Thus, it will be useful to have a measure that quantifies the dissimilarity between two representations, indicating how far they are from being equivalent. This measure should be minimized when the representations are equivalent. We do not present such a metric for the general case. In our experiments, we will want to compare learned encoders with a deterministic encoder corresponding to the ground truth $\underline{T}$ for the case $\beta \to +\infty$. Hence, a metric that captures equivalence between a representation $T$ and a deterministic representation $\underline{T}$ will suffice. The variation of information (Meilă, 2003; 2007) $\mathrm{VI}(T_1; T_2) \coloneqq H(T_1 \mid T_2) + H(T_2 \mid T_1)$ is such a metric: $\mathrm{VI}(T, \underline{T}) = 0$ exactly when $T$ and $\underline{T}$ are equivalent (see Proposition 20).

## 8 EXPERIMENTAL RESULTS

In our experiments, we aim to minimize a reparameterized version of the CIB, given by $\mathcal{L}^{\gamma}_{\text{CIB}}[q_{T|X}] := (1 - \gamma)I(X;T) - \gamma I_c(Y \mid do(T))$, where $\gamma \in [0, 1]$ is the trade-off parameter. By a slight abuse of notation, we distinguish this from the original parameterization of the CIB solely by the superscript $\gamma$. The parameter $\gamma$ has a more intuitive interpretation than $\beta$, representing the fraction of the CIB that the interventional sufficiency term accounts for. Additionally, it simplifies hyperparameter searches when using optimization algorithms, since the magnitude of the values of $\mathcal{L}^{\gamma}_{\text{CIB}}$ remains relatively stable with variations in $\gamma$, unlike what happens with $\mathcal{L}^{\beta}_{\text{CIB}}$ and $\beta$. It is straightforward to verify that minimizing the CIB $\mathcal{L}^{\beta}_{\text{CIB}}$ is equivalent to minimizing its reparameterization $\mathcal{L}^{\gamma}_{\text{CIB}}$, provided that $\gamma$ is selected appropriately (see Proposition 18).

In this section, we will demonstrate the application of the CIB Lagrangian by finding its minimum for three problems of increasing complexity and for different values of $\gamma$, using the local search algorithms[2] (Simplex) Projected Gradient Descent (pGD) and (Simplex) Projected Simulated Annealing Gradient Descent (pSAGD) described in Appendix E, and checking whether the results are what we expect.

Specifically, in each experiment we will check whether:

(a) If $\gamma = 1$, the learned representation $T$ coincides (modulo $\cong$) with the ground truth $\underline{T}$ of that experiment. That is, whether $\text{VI}(T, \underline{T}) = 0$.

(b) If $\gamma = 0$, the learned representation has $I(X;T) = 0$.

(c) Larger $\gamma$ values correspond to larger (or at least not smaller) values of $I_c(Y \mid do(T))$. That is, if $\gamma_1 > \gamma_2$, one has that the causal information gain for the encoder learned when $\gamma = \gamma_1$ is larger or equal to that of the encoder learned when $\gamma = \gamma_2$.

If (a), (b), and (c) hold, this provides evidence that the CIB can be used to learn representations that maximize control (by setting $\gamma = 1$), maximize compression (by setting $\gamma = 0$), or strike a balance between the two (by setting $\gamma \in (0, 1)$). This is also evidence that the proposed local search algorithms succeed in optimizing the CIB objective. Note that ground truth is only available for $\gamma = 1$ and $\gamma = 0$. For $\gamma = 1$, the optimal solution will be clear for the proposed case studies. For $\gamma = 0$, the solution should be maximally compressive, regardless of causal control over $Y$. For $\gamma \notin \{0, 1\}$, there is no obvious ground truth, but the reasonableness of the results can still be assessed as described in (c). It is also noteworthy that encoders trained with different $\gamma$ values may achieve the same $I_c(Y \mid do(T))$, although increasing $\gamma$ should not decrease $I_c(Y \mid do(T))$. This property is analogous to the Information Bottleneck (IB) framework, where distinct $\beta$ values often yield the same sufficiency value (Kolchinsky et al., 2018).

### 8.1 LEARNING ODD AND EVEN

**Setup** Consider the SCM in Figure 2a, which represents a scenario where the parity of $X$ determines the outcome $Y$ with some uncertainty, parameterized by $u_Y$. To preserve the control that $X$ has over $Y$, a representation $T$ of $X$ should reflect the parity of $X$. Consequently, when $T$ is binary and we aim to maximize the causal control of $T$ over $Y$, $T$ must be equivalent to $\underline{T} = X \mod 2$. This will serve as the ground truth (modulo $\cong$) for the case $\gamma = 1$.

**Results** For each $\gamma$ value, we employ an ensemble of $4$ pGD optimizers with learning rates $1.0$ and $0.1$. As shown in Figure 1a, the experiment satisfies all checks (a), (b), and (c).

This case is straightforward, with $T$ abstracting a single variable and no confounders. The backdoor criterion is trivially satisfied by the empty set, and the Conditional Information Bottleneck (CIB) reduces to the standard Information Bottleneck (IB). We now consider a more complex case where controlling for confounding is crucial.

---

[2]The learning rates were chosen based on hyperparameter searches for the $\gamma = 1$ cases as well as the typical values of the gradient norm.

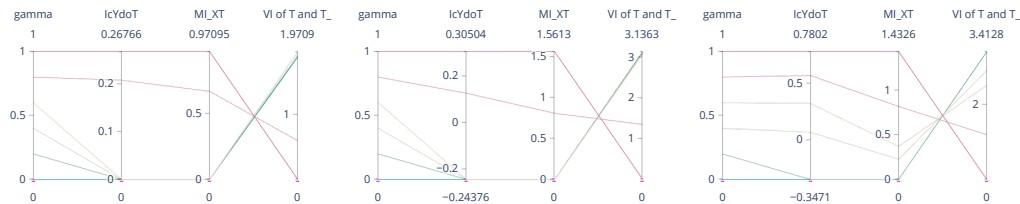

(a) Odd and Even experiment with $u_Y = 0.2$.

(b) Confounded Addition experiment with $r_Y = 0.5$.

(c) Genetic Mutations experiment with $b_{X_i} = 0.3$, $b_Y = 0.1$ and $b_S = 0.5$.

Figure 1: Experimental results. Each line corresponds to the representation found for the chosen $\gamma \in \{0, 0.2, \ldots, 1.0\}$. The axis labels `gamma`, `IcYdoT`, `MI_XT` and `VI of T and T_` correspond to $\gamma$, $I_c(Y \mid do(T))$, $I(X; T)$ and $\mathrm{VI}(T, \underline{T})$, respectively. Maximal $\gamma$ leads to maximal causal control and learning the ground truth, while minimal $\gamma$ leads to maximal compression, as expected. Larger $\gamma$ values result in larger (or at least not smaller) causal control. Notice that some lines overlap.

## 8.2 LEARNING ADDITION IN THE PRESENCE OF STRONG CONFOUNDING

**Setup** Consider the SCM in Figure 2b, which represents a situation where $Y$ is controlled by $X = (X_1, X_2)$ through the sum $X_1 + X_2 \in \{0, 1, 2\}$, and $W$ confounds $X_1$ and $Y$. Notice that $W$ satisfies the backdoor criterion relative to $(X, Y)$. To preserve the control that $X$ has over $Y$, a representation $T$ of $X$ should keep the value of $X_1 + X_2 \in \{0, 1, 2\}$. This is because, by construction of the Structural Causal Model (SCM), the sum of $X_1$ and $X_2$ is the only aspect of $X$ that can be manipulated to affect $Y$. Therefore, if $T$ is chosen to be a 3-valued variable and one aims to maximize causal control of $T$ over $Y$, it follows that $T$ should be equivalent to $\underline{T} = X_1 + X_2$. This will be the ground truth abstraction (modulo $\cong$) for the case $\gamma = 1$.

**Results** For each value of $\gamma$, we employ an ensemble of 6 pSAGD optimizers with a temperature of 10.0, learning rates 1.0 and 10.0. As shown in Figure 1b, the experiment satisfies all checks (a), (b), and (c). This shows in particular that our method successfully deals with the confounding effect of $W$.

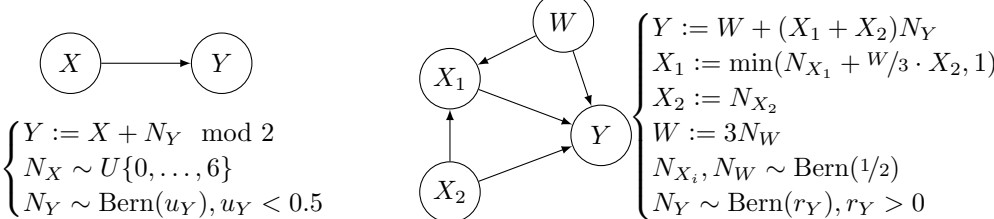

(a) SCM for the Odd and Even experiment. We use $u_Y = 0.2$.

(b) SCM for the Confounded Addition experiment. We use $r_Y = 0.5$. Notice that $Y \in \{0, 1, 2, 3, 4, 5\}$ and $W \in \{0, 3\}$.

Figure 2: SCMs for two experiments.

## 8.3 GENETIC MUTATIONS

**Setup** Consider the SCM depicted in Figure 3, which represents a scenario of genetic mutations in mice like the one described in Section 1. Notice that $S$ satisfies the backdoor criterion relative to $(X, Y)$. By inspecting the structural assignments in Figure 3, we can see that the mutations interact in a complex, non-additive way with respect to the body mass $Y$. Specifically, individual mutations at $s_1$ or $s_2$ have an equivalent, relatively small effect on $Y$, while having both mutations simultaneously would have a profound impact, larger than the sum of the impacts of the individual mutations. Furthermore, having a mutation at $s_3$ has no effect on its own, but partially protects against the effect of simultaneous mutations at $s_1$ and $s_2$. Finally, $X_4$ has no effect on $Y$ whatsoever. This

means in particular that $T$ should be able to distinguish cases where both $s_1$ and $s_2$ are mutated but $s_3$ is not, and where all three are mutated.

This setting is characterized by an increased number of variables, confounding, and complex epistatic interactions between variables, which the learned encoder must capture. Furthermore, the encoder should recognize that the maximal control solution does not need to to account for one of the variables in $X$ (specifically, $X_4$). Denote by $\theta(X)$ the sum $X_1 + X_2 + (2 - X_3) \cdot X_1 \cdot X_2$ of all the terms in the structural assignment of $Y$ which involve $X$. This expression represents the aspect of $Y$ that $X$ can influence, and thus it is the information that $T$ should capture to achieve maximal control. It follows that a 4-valued representation learned with $\gamma = 1$ case should be equivalent to the representation $\underline{T}$ with encoder $q_{T|X}$ given by $q_{\underline{T}|X}(t \mid x_1, x_2, x_3, x_4) = \delta_{t, \theta(x_1, x_2, x_3, x_4)}$. This will be the ground truth abstraction (modulo $\cong$) for the case $\gamma = 1$.

**Results**  For each value of $\gamma$, we employ an ensemble of 12 pSAGD optimizers with a temperature of 10.0, learning rates $10^0$, $10^1$, $10^2$ and $10^4$. As shown in Figure 1c, the experiment satisfies all checks (a), (b), and (c). This shows that our method can capture complex interaction effects between the variables, while simultaneously dealing with the confounding effect of $S$.

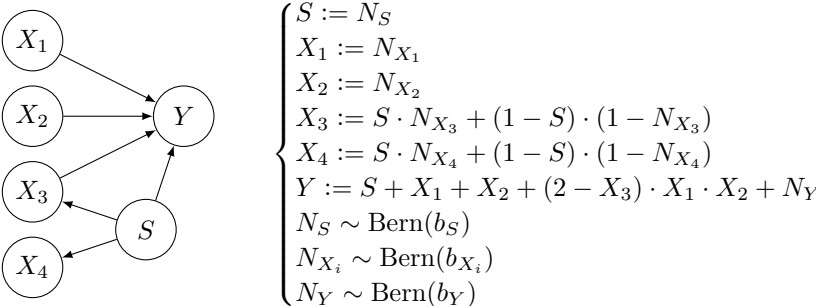

Figure 3: SCM for the Genetic Mutations experiment. This models the running example introduced in Section 1, so that the $X_i$ indicate the existence of mutations at certain locations in a mouse gene, $S$ is the population strain, and $Y$ is the mouse body mass. We fix $b_{X_i} = 0.3$, $b_Y = 0.1$ and $b_S = 0.5$.

## 9 RELATED WORK

There has been other work investigating a problem related to our search for an optimal causal representation. In Chalupka et al. (2017; 2014; 2016b;a)), the authors search for "causal macrovariables" of higher-dimensional "microvariables", described as coarse representations of the microvariables $X$ and $Y$ which preserve the causal relation between them[3]. Their definition of macrovariable is distinct from our definition of optimal causal representations. Namely, it is not based on information-theoretical concepts such as compression and sufficiency, but on clustering together the values of $X$ resulting in the same post-intervention distributions over $Y$. It would be interesting to see in what cases our approach coincides with theirs. Note that the two approaches cannot align in general, since their method does not incorporate causality, as their clustering procedure relies on conditional distributions, which, as they demonstrate, necessarily yield a finer-grained representation than truly causal clusters. Moreover, our abstractions $T$ can, at least for $\gamma \neq 1$, be stochastic mappings of $X$, while their macrovariables are deterministic mappings of $X$. Höltgen (2021); Jammalamadaka et al. (2023) tried to address the problem of finding causal macrovariables using the standard information bottleneck framework. Specifically, they use variational autoencoders (VAEs), relying on the fact that $\beta$-VAEs (when used for the task of reconstruction) minimize the information bottleneck (Achille & Soatto, 2018). However, their method does not account for causality, being limited to scenarios without confounding due to its reliance on the standard IB.

The study of causal abstractions has been conducted from yet another, although related, point of view. Namely, there have been efforts to formalize when exactly a given causal model can be seen as

---

[3]In general, they search for representations not only of $X$ but also of $Y$. We are only interested in how they construct representations of $X$.

a causally consistent abstraction of another one (Rubenstein et al., 2017; Beckers & Halpern, 2019). The core idea is that intervening on a variable in the low-level model and then transitioning to the high-level model should yield the same distribution over the model variables as first transitioning to the high-level model and then intervening. To account for cases where this commutativity between abstracting and intervening is imperfect, Beckers et al. (2020) introduced approximate abstractions. Although these works do not provide methods for constructing causal model abstractions, recent efforts have focused on learning such high-level models. Notably, Xia & Bareinboim (2024) present a method for constructing high-level models based on clustering low-level variables, while Zennaro et al. (2023) and Felekis et al. (2024) focus on learning an abstraction map given both low-level and high-level SCMs, each by minimizing a distinct measure of abstraction error. In contrast, our method does not learn a high-level SCM, but a representation of $X$ which is causally relevant. It seems to us that a possibly fruitful avenue of research consists of studying whether representation learning method introduced in this paper can be used to construct high-level causal models. In fact, the connection between causal representation learning and the task of learning abstractions of causal models has been noted before (Zennaro et al., 2023). Furthermore, the macrovariables from Chalupka et al. (2017) were already shown to induce high-level causal models that satisfy the definition of approximate abstraction (Beckers et al., 2020), suggesting that the causal representations learned using the CIB method may play a similar role.

## 10    CONCLUSION AND FUTURE WORK

We extended the notion of optimal representation to the causal setting, resulting in an axiomatic characterization of optimal causal representations. Just as the information bottleneck (IB) method can be used to learn optimal representations, so can the causal information bottleneck (CIB) method introduced in this paper be used to learn optimal causal representations. The CIB, which depends on the interventional distributions $p(y \mid do(t))$, needs to be computed during the learning procedure, which consists of solving a constrained minimization problem. The exact expression for the CIB will depend on the causal structure of the system under study. We focused on cases where there is a set $\mathbf{Z}$ satisfying the backdoor criterion relative to $(X, Y)$. This allows us to derive a backdoor adjustment formula for $p(y \mid do(t))$, and thus successfully apply a minimization algorithm to minimize the CIB. Specifically, we introduced a local search algorithm, referred to as projected simulated annealing gradient descent (pSAGD), which integrates simulated annealing and gradient descent techniques with a projection operator to maintain constraint satisfaction throughout the minimization process. In order to compare different representations learned by our algorithm, we introduced a novel notion of equivalence of representations, which partitions representations into equivalence classes, termed abstractions, and showed that the variation of information can be used to assess whether two representations are equivalent, as long as one of them is deterministic. We experimentally validated that the learned representations in three toy models of increasing complexity align with our expectations.

Future research directions include exploring alternative methods for incorporating causality into the information bottleneck framework, such as focusing on causal properties other than causal control, like proportionality (Pocheville et al., 2015). Our approach can also be extended to scenarios where the backdoor criterion does not hold by leveraging do-calculus, allowing for the automatic computation of post-intervention distributions for interventions on representations. Another area worth investigating relates to fine-tuning the trade-off between compression and interventional sufficiency. Kolchinsky et al. (2018) highlight that, in the context of the Information Bottleneck (IB), different values of $\beta$ can often result in the same sufficiency. Similarly, in our experiments we observed that different values of $\gamma$ (and thus $\beta$) often produced the same interventional sufficiency values. Future work could explore strategies similar to those used by Kolchinsky et al. (2018) to address this. Additionally, another natural next step would be to adapt the causal information bottleneck (CIB) method to continuous variables, for example by using variational autoencoders (Kingma & Welling, 2013) to minimize the CIB Lagrangian, as was previously done for the standard IB (Alemi et al., 2016). Finally, one can explore the relationship between our representation learning method and the framework of causal abstractions, similar to how Beckers et al. (2020) connect the latter with the approach from Chalupka et al. (2017).

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

## A MORE PRELIMINARIES

### A.1 ENTROPY AND MUTUAL INFORMATION

In this subsection we will state the definitions of entropy, conditional entropy and mutual information. In the interest of space, we will not try to motivate these definitions. For more, see Cover & Thomas (2006).

**Definition 10** (Entropy and Cond. Entropy (Cover & Thomas, 2006)). *Let $X$ be a discrete random variable with range $R_X$ and $p$ be a probability distribution for $X$. The* entropy *of $X$ w.r.t. the distribution $p$ is*[4]

$$H_{X \sim p}(X) := - \sum_{x \in R_X} p(x) \log p(x). \tag{7}$$

*Entropy is measured in* bit. *If the context suggests a canonical probability distribution for $X$, one can write $H(X)$ and refers to it simply as the* entropy *of $X$.*
*The* conditional entropy *$H(Y \mid X)$ of $Y$ conditioned on $X$ is the expected value w.r.t. $p_X$ of the entropy $H(Y \mid X = x) := H_{Y \sim p_{Y \mid X = x}}(Y)$:*

$$H(Y \mid X) := \mathbb{E}_{x \sim p_X} [H(Y \mid X = x)]. \tag{8}$$

This means that the conditional entropy $H(Y \mid X)$ is the entropy of $H(Y)$ that remains on average if one conditions on $X$.

There are two common equivalent ways to define mutual information (often called information gain).

**Definition 11** (Mutual Information and Cond. Mutual Information (Cover & Thomas, 2006)). *Let $X$ and $Y$ be discrete random variables with ranges $R_X$ and $R_Y$ and distributions $p_X$ and $p_Y$, respectively. The* mutual information *between $X$ and $Y$ is*

$$I(X;Y) := \sum_{x,y \in R_X \times R_Y} p_{X,Y}(x,y) \log \frac{p_{X,Y}(x,y)}{p_X(x) p_Y(y)}. \tag{9}$$

*Or equivalently:*

$$\begin{aligned} I(X;Y) &:= H(Y) - H(Y \mid X) \\ &= H(X) - H(X \mid Y). \end{aligned} \tag{10}$$

*Let $Z$ be another discrete random variable. The* conditional mutual information *between $X$ and $Y$ conditioned on $Z$ is:*

$$\begin{aligned} I(X;Y \mid Z) &:= H(Y \mid Z) - H(Y \mid X, Z) \\ &= H(X \mid Z) - H(X \mid Y, Z). \end{aligned} \tag{11}$$

The view of mutual information as entropy reduction from Equation (10) is the starting point for the definition of causal information gain.

### A.2 MORE ON CAUSAL ENTROPY AND CAUSAL INFORMATION GAIN

In this section, we will define causal entropy and causal information gain. The latter will be an essential component of our method. See Simoes et al. (2023) for a thorough discussion about these concepts. Let $X$ and $Y$ be endogenous variables of an SCM $\mathfrak{C}$. The causal entropy of $Y$ for $X$ is the entropy of $Y$ that remains, on average, after one atomically intervenes on $X$. Its definition is analogous to that of conditional entropy (see Definition 10). Concretely, causal entropy is the average uncertainty one has about $Y$ if one sets $X$ to $x$ with probability $p_{X^\star}(x)$, where $p_{X^\star}$ specifies the distribution over interventions.

**Definition 12** (Causal Entropy, $H_c$ (Simoes et al., 2023)). *Let $Y$, $X$ and $X^\star$ be random variables such that $X$ and $X^\star$ have the same range and $X^\star$ is independent of all variables in $\mathfrak{C}$. We say that $X^\star$ is an* intervention protocol *for $X$. The* causal entropy *$H_c(Y \mid do(X \sim X^\star))$ of $Y$ for $X$ given the intervention protocol $X^\star$ is the expected value w.r.t. $p_{X^\star}$ of the entropy $H(Y \mid do(X = x)) := H_{Y \sim p_Y^{do(X=x)}}(Y)$ of the interventional distribution $p_Y^{do(X=x)}$. That is:*

$$H_c(Y \mid do(X \sim X^\star)) := \mathbb{E}_{x \sim p_{X^\star}} [H(Y \mid do(X = x))]. \tag{12}$$

---

[4]In this article, $\log$ denotes the logarithm to the base 2.

Causal information gain extends mutual information/information gain to the causal context. While mutual information between two variables $X$ and $Y$ is the average reduction in uncertainty about $Y$ if one observes the value of $X$ (see Equation (10)), the causal information gain of $Y$ for $X$ is the average decrease in the entropy of $Y$ after one atomically intervenes on $X$ (following an intervention protocol $X^\star$).

**Definition 13** (Causal Information Gain, $I_c$ (Simoes et al., 2023))**.** *Let $Y$, $X$ and $X^\star$ be random variables such that $X^\star$ is an intervention protocol for $X$. The* causal information gain *$I_c(Y \mid do(X \sim X^\star))$ of $Y$ for $X$ given the intervention protocol $X^\star$ is the difference between the entropy of $Y$ w.r.t. its prior and the causal entropy of $Y$ for $X$ given the intervention protocol $X^\star$. That is:*

$$I_c(Y \mid do(X \sim X^\star)) := H(Y) - H_c(Y \mid do(X \sim X^\star)). \tag{13}$$

The causal information gain of $Y$ for $X$ was proposed in Simoes et al. (2023) as a measure of the "(causal) control that variable $X$ has over the variable $Y$", that is, the reduction of uncertainty about $Y$ that results from intervening on $X$. It is noteworthy that $I_c(Y \mid do(X))$ can be negative, in contrast with mutual information.

As described in Section 5, we choose a uniform prior over the interventions on $X$; that is, a uniform protocol $p_{X^\star} = p^\star$ over the low-level variables $X$. This induces an intervention protocol $p^\star(t) = \sum_x q(t \mid x) p^\star(x)$ for $T$. To simplify notation, we omit the protocol and write simply $I_c(Y \mid do(X))$ and $I_c(Y \mid do(T))$.

### A.3 MORE ON STRUCTURAL CAUSAL MODELS

One can model the causal structure of a system by means of a "structural causal model", which can be seen as a Bayesian network (Koller & Friedman, 2009) whose graph $G$ has a causal interpretation and each conditional probability distribution (CPD) $P(X_i \mid \text{Pa}_{X_i})$ of the Bayesian network stems from a deterministic function $f_{X_i}$ (called "structural assignment") of the parents of $X_i$. In this context, it is common to separate the parent-less random variables (which are called "exogenous" or "noise" variables) from the rest (called "endogenous" variables). Only the endogenous variables are represented in the structural causal model graph. As is commonly done (Peters et al., 2017), we assume that the noise variables are jointly independent and that exactly one noise variable $N_{X_i}$ appears as an argument in the structural assignment $f_{X_i}$ of $X_i$. In full rigor[5](Peters et al., 2017):

**Definition 14** (Structural Causal Model)**.** *Let $X$ be a random variable with range $R_X$ and $\mathbf{W}$ a random vector with range $R_{\mathbf{W}}$. A* structural assignment for $X$ from $\mathbf{W}$ *is a function $f_X \colon R_{\mathbf{W}} \to R_X$. A* structural causal model (SCM) *$\mathfrak{C} = (\mathbf{X}, \mathbf{N}, S, p_{\mathbf{N}})$ consists of:*

1. *A random vector $\mathbf{X} = (X_1, \dots, X_n)$ whose variables we call* endogenous.

2. *A random vector $\mathbf{N} = (N_{X_1}, \dots, N_{X_n})$ whose variables we call* exogenous *or* noise.

3. *A set $S$ of $n$ structural assignments $f_{X_i}$ for $X_i$ from $(\text{Pa}_{X_i}, N_{X_i})$, where $\text{Pa}_{X_i} \subseteq \mathbf{X}$ are called* parents *of $X_i$. The* causal graph *$G^{\mathfrak{C}} := (\mathbf{X}, E)$ of $\mathfrak{C}$ has as its edge set $E = \{(P, X_i) : X_i \in \mathbf{X}, \ P \in \text{Pa}_{X_i}\}$. The $\text{Pa}_{X_i}$ must be such that the $G^{\mathfrak{C}}$ is a directed acyclic graph (DAG).*

4. *A jointly independent probability distribution $p_{\mathbf{N}}$ over the noise variables. We call it simply the* noise distribution.

We denote by $\mathfrak{C}(\mathbf{X})$ the set of SCMs with vector of endogenous variables $\mathbf{X}$. Notice that for a given SCM the noise variables have a known distribution $p_{\mathbf{N}}$ and the endogenous variables can be written as functions of the noise variables. Therefore the distributions of the endogenous variables are themselves determined if one fixes the SCM. This brings us to the notion of the entailed distribution[5] (Peters et al., 2017):

**Definition 15** (Entailed distribution)**.** *Let $\mathfrak{C} = (\mathbf{X}, \mathbf{N}, S, p_{\mathbf{N}})$ be an SCM. Its* entailed distribution $p_{\mathbf{X}}^{\mathfrak{C}}$ *is the unique joint distribution over $\mathbf{X}$ such that $\forall X_i \in \mathbf{X}$, $X_i = f_{X_i}(\text{Pa}_{X_i}, N_{X_i})$. It is often simply denoted by $p^{\mathfrak{C}}$. Let $\mathbf{x}_{-i} := (x_1, \dots, x_{i-1}, x_{i+1}, \dots, x_n)$. For a given $X_i \in \mathbf{X}$, the* marginalized

---

[5]We slightly rephrase the definition provided in Peters et al. (2017) for our purposes.

*distribution $p_{X_i}^{\mathfrak{C}}$ given by $p_{X_i}^{\mathfrak{C}}(x_i) = \sum_{\mathbf{x}_{-i}} p_{\mathbf{X}}^{\mathfrak{C}}(\mathbf{x})$ is also referred to as* entailed distribution *(of $X_i$).*

An SCM allows us to model interventions on the system. The idea is that an SCM represents how the values of the random variables are generated, and by intervening on a variable we are effectively changing its generating process. Thus intervening on a variable can be modeled by modifying the structural assignment of said variable, resulting in a new SCM differing from the original only in the structural assignment of the intervened variable, and possibly introducing a new noise variable for it, in place of the old one. Naturally, the new SCM will have an entailed distribution which is in general different from the distribution entailed by the original SCM.

The most common type of interventions are the so-called "atomic interventions", where one sets a variable to a chosen value, effectively replacing the distribution of the intervened variable with a point mass distribution. In particular, this means that the intervened variable has no parents after the intervention. This is the only type of intervention that we will need to consider in this work. Formally[5] Peters et al. (2017):

**Definition 16** (Atomic intervention). *Let $\mathfrak{C} = (\mathbf{X}, \mathbf{N}, S, p_{\mathbf{N}})$ be an SCM, $X_i \in \mathbf{X}$ and $x \in R_{X_i}$. The* atomic intervention $do(X_i = x)$ *is the function $\mathfrak{C}(\mathbf{X}) \to \mathfrak{C}(\mathbf{X})$ given by $\mathfrak{C} \mapsto \mathfrak{C}^{do(X_i=x)}$, where $\mathfrak{C}^{do(X_i=x)}$ is the SCM that differs from $\mathfrak{C}$ only in that the structural assignment $f_{X_i}(\mathrm{Pa}_{X_i}, N_{X_i})$ is replaced by the structural assignment $\tilde{f}_{X_i}(\tilde{N}_{X_i}) = \tilde{N}_{X_i}$, where $\tilde{N}_{X_i}$ is a random variable with range $R_{X_i}$ and[6] $p_{\tilde{N}_{X_i}}(x_i) = \mathbf{1}_x(x_i)$ for all $x_i \in R_{X_i}$. Such SCM is called the* post-atomic-intervention SCM. *One says that the variable $X_i$ was* (atomically) intervened on. *The distribution $p^{do(X_i=x)} := p^{\mathfrak{C}^{do(X_i=x)}}$ entailed by $\mathfrak{C}^{do(X_i=x)}$ is called the* post-intervention distribution *(w.r.t. the atomic intervention $do(X_i = x)$ on $\mathfrak{C}$).*

# B SUPPLEMENTARY MATERIAL ON THE CIB

## B.1 LAGRANGE MULTIPLIERS AND THE CIB

*Remark* 17 (Distinctions from classical Lagrange multipliers). The minimization problem in Equation (2) involves both equality and inequality constraints. To tackle this problem using the method of Lagrange multipliers (Nocedal, 2006) directly, we would need to construct a Lagrangian of the form $\mathcal{L}(q_{T|X}, \beta, (\lambda_x)_x, (\mu_{x,t})_{x,t}) = I(X;T) - \beta(I_c(Y \mid do(T)) - D) - \sum_x \lambda_x g_x((q_{T|X=x})_x) - \sum_{x,t} \mu_{x,t} h_{x,t}(q_{T=t|X=x})$, where the $g_x$ are the restriction functions ensuring that all the conditional distributions $q_{T|X=x}$ are normalized and the $h_{x,t}$ are the inequality restriction functions ensuring non-negativity of the conditional distributions $q_{T|X=x}$. Hence the last two terms of $\mathcal{L}$, along with the appropriate Karush-Kuhn-Tucker (KKT) conditions, would guarantee that the conditional distributions $q_{T|X=x}$ belong to the simplex $\Delta^{|R_T|-1}$. However, finding the stationary points of the Lagrangian with respect to all its arguments, *i.e.*, those where $\nabla_{q_{T|X},(\lambda_x)_x,(\mu_{x,t})}\mathcal{L} = 0$, would be a formidable task. Instead, we follow the approach of Tishby et al. (2000); Strouse & Schwab (2017) and impose the simplex constraint separately, outside of the Lagrangian multipliers method, leaving us only with the sufficiency constraint. Furthermore, in contrast with the classic method of Lagrangian multipliers, the multiplier $\beta$ is fixed, so that $D$ is not chosen directly, but only indirectly through the choice of $\beta$.

## B.2 WEIGHTED CAUSAL INFORMATION BOTTLENECK LAGRANGIAN

In this section, it will be useful to distinguish $\mathcal{L}_{\mathrm{CIB}}^{\gamma}$ from the original CIB $\mathcal{L}_{\mathrm{CIB}}^{\beta}$. We call the former the *weighted causal information bottleneck (wCIB) Lagrangian* and denote it $\mathcal{L}_{\mathrm{wCIB}}^{\gamma}$.

**Proposition 18.** *Let $\beta \in \mathbb{R}^+$ and $\gamma = \frac{\beta}{1+\beta}$. Then the minimizers of $\mathcal{L}_{\mathrm{CIB}}^{\beta}[q_{T|X}]$ are the same as those of $\mathcal{L}_{\mathrm{wCIB}}^{\gamma}[q_{T|X}]$.*

---

[6]We denote by $\mathbf{1}_x$ the indicator function of $x$, so that $\mathbf{1}_x(x_i) = \begin{cases} 1, & x_i = x \\ 0, & \text{otherwise} \end{cases}$.

*Proof.*

$$\mathcal{L}_{\mathrm{wCIB}}^{\gamma} = (1 - \gamma)I(Y; X) - \gamma I_c(Y \mid do(T))$$

$$= \frac{1}{1 + \beta}I(Y; X) - \frac{\beta}{1 + \beta}I_c(Y \mid do(T)) \tag{14}$$

$$= \frac{1}{1 + \beta}\mathcal{L}_{\mathrm{CIB}}^{\beta}.$$

Therefore, the wCIB with the chosen $\gamma$ is simply a rescaling of the CIB with scaling factor $\frac{1}{1+\beta} \in (0, 1]$. Since this factor is always positive, it follows that $\mathcal{L}_{\mathrm{wCIB}}^{\gamma}$ and $\mathcal{L}_{\mathrm{CIB}}^{\beta}$ attain their minima at the same points. $\square$

Notice that $\beta = \frac{\gamma}{1-\gamma}$ for $\gamma < 1$, and that $\beta \to +\infty$ as $\gamma \to 1$. In case maximal causal control of $T$ is desired without consideration of compression, we can use the wCIB with $\gamma = 1$, in which case we formally set $\beta = +\infty$.

## C SUPPLEMENTARY MATERIAL ON INTERVENTIONS ON REPRESENTATIONS

### C.1 EFFECT OF A REPRESENTATION INTERVENTION ON THE TARGET VARIABLE

For $Y \in \mathbf{V}$, one has that

$$p_Y^{do(T=t)}(y) = p(y \mid do(T = t)) = \sum_{v_1,\dots,v_m,\tilde{x}} p(v_1, \dots, v_m, \tilde{x}, y \mid do(T = t))$$

$$= \sum_x p^{\star}(x \mid t) \sum_{v_1,\dots,v_m,\tilde{x}} p(v_1, \dots, v_m, \tilde{x}, y \mid do(X = x)) \tag{15}$$

$$= \sum_x p^{\star}(x \mid t)p(y \mid do(X = x)),$$

where $V_1, \dots, V_m$ are all the variables in $\mathbf{V}$ except for $X$ and $Y$.

### C.2 PROOF OF THE BACKDOOR ADJUSTMENT FORMULA FOR CAUSAL REPRESENTATIONS

*Proof of Proposition 6.*

$$p(y \mid do(t)) = \sum_x p^{\star}(x \mid t)p^{\mathfrak{C};do(X=x)}(y)$$

$$= \sum_x p^{\star}(x \mid t) \sum_z p^{\mathfrak{C}}(z)p^{\mathfrak{C}}(y \mid x, z) \tag{16}$$

$$= \sum_z p^{\mathfrak{C}}(z) \sum_z p^{\mathfrak{C}}(y \mid x, z)\frac{q(t \mid x)p^{\star}(x)}{\sum_{\dot{x}} q(t \mid \dot{x})p^{\star}(\dot{x})},$$

where the second equality follows from the backdoor adjustment formula (Pearl, 2009), and the last one from the definition of the intervention decoder. $\square$

## D SUPPLEMENTARY MATERIAL ON COMPARING REPRESENTATIONS

**Proposition 19.** *The relation $\cong$ of equivalence of abstractions is an equivalence relation.*

*Proof.* Reflexivity of $\cong$ is immediate: just take $\sigma = \mathrm{id}$. We now show that $\cong$ is symmetric. Assume that $T_1 \cong T_2$, with corresponding bijection $\sigma$. Denote by $\sigma^{-1}$ its inverse. Let $t_2 \in \mathrm{supp}(T_2)$ and $t_1 = \sigma^{-1}(t_2)$. Then,

$$q_{T_2|X}(t_2 \mid x) = q_{T_2|X}(\sigma(t_1) \mid x)$$

$$= q_{T_1|X}(t_1 \mid x) \tag{17}$$

$$= q_{T_1|X}(\sigma^{-1}(t_2) \mid x).$$

This shows that $T_2 \cong T_1$. Finally, we show transitivity. Let $T_1$, $T_2$ and $T_3$ be representations of $X$ such that $T_1 \cong T_2$ and $T_2 \cong T_3$, with bijections $\sigma_{12}$ and $\sigma_{23}$. Then,

$$
\begin{aligned}
q_{T_1|X}(t_1 \mid x) &= q_{T_2|X}(\sigma_{12}(t_1) \mid x) \\
&= q_{T_3|X}(\sigma_{23}(\sigma_{12}(t_1)) \mid x).
\end{aligned}
\tag{18}
$$

Hence $T_1 \cong T_3$ with bijection $\sigma_{23} \circ \sigma_{12}$. $\qquad\square$

**Proposition 20.** *Let $T_1$ and $T_2$ be representations of $X$. If $\mathrm{VI}(T_1, T_2) = 0$, then $T_1 \cong T_2$. Furthermore, the converse also holds if $T_1$ is a deterministic representation of $X$.*

*Proof.* Note that $\mathrm{VI}(T_1, T_2) = 0$ if and only if $H(T_2 \mid T_1) = H(T_1 \mid T_2) = 0$. Recall that $H(T_2 \mid T_1) = -\sum_{t_1 \in \mathrm{supp}(T_1)} p(t_1) \sum_{t_2 \in \mathrm{supp}(p_{T_2|T_1})} p(t_2 \mid t_1) \log p(t_2 \mid t_1)$, which is zero if and only if $p(t_2 \mid t_1) \in \{0, 1\}$ for all $t_1, t_2$ in the respective supports. By the same token, $p(t_1 \mid t_2) \in \{0, 1\}$. Define $\sigma \colon \mathrm{supp}(T_1) \to \mathrm{supp}(T_2)$ by $\sigma(t_1) = \arg_{t_2}(p(t_2 \mid t_1) = 1)$. Similarly, define $\sigma^{-1} \colon \mathrm{supp}(T_2) \to \mathrm{supp}(T_1)$ by $\sigma^{-1}(t_2) = \arg_{t_1}(p(t_1 \mid t_2) = 1)$. Then, $\sigma^{-1}$ is the inverse of $\sigma$. To see this, note that, for all $t_2^* \in \mathrm{supp}(T_2)$, one has $\sigma(\sigma^{-1}(t_2^*)) = \arg_{t_2}(p(t_2 \mid \sigma^{-1}(t_2^*)) = 1)$. But $p(t_2^* \mid \sigma^{-1}(t_2^*)) = \frac{p(\sigma^{-1}(t_2^*)|t_2^*)p(t_2^*)}{p(\sigma^{-1}(t_2^*))} = \frac{1 \cdot p(t_2^*)}{p(\sigma^{-1}(t_2^*))}$, which cannot be zero. Since $p(t_2 \mid t_1) \in \{0, 1\}$ for all $t_1, t_2$, one then has that $p(t_2^* \mid \sigma^{-1}(t_2^*)) = 1$, and therefore $\sigma(\sigma^{-1}(t_2^*)) = t_2^*$. Now, let $t_1 \in \mathrm{supp}(T_1)$ and $t_2 = \sigma(t_1)$. Then $q_{T_1|X}(t_1 \mid x) = q_{T_1|X}(\sigma^{-1}(t_2) \mid x) = q_{T_2|X}(t_2 \mid x) = q_{T_2|X}(\sigma(t_1) \mid x)$. Hence $T_1 \cong T_2$.

Assume now that $T_1$ is a deterministic representation of $X$, *i.e.*, $q_{T_1|X}(t_1 \mid x) \in \{0, 1\}$ for all $t_1 \in R_T, x \in \mathrm{supp}(X)$. Assume further that $T_1 \cong T_2$ with bijection $\sigma$. Then clearly $T_2$ is also deterministic. Making use of the definition of representation, we have $p(t_2 \mid t_1) = \sum_x p(t_2 \mid x)p(x \mid t_1)$. If there is no $x$ for which $p(t_2 \mid x) = 1$, then this is zero. Otherwise, $p(t_2 \mid t_1) = \sum_x p(t_2 \mid x)\frac{p(t_1|x)p(x)}{\sum_{\dot{x}} p(t_1|\dot{x})p(\dot{x})} = \sum_x q_{T_2|X}(t_2 \mid x)\frac{q_{T_1|X}(t_1|x)p(x)}{\sum_{\dot{x}} q_{T_1|X}(t_1|\dot{x})p(\dot{x})} = \sum_{x \in S} \frac{q_{T_1|X}(t_1|x)p(x)}{\sum_{\dot{x}} q_{T_1|X}(t_1|\dot{x})p(\dot{x})} = \sum_{x \in S} \frac{q_{T_2|X}(\sigma(t_1)|x)p(x)}{\sum_{\dot{x}} q_{T_2|X}(\sigma(t_1)|\dot{x})p(\dot{x})}$, where $S$ is the set of values $x$ where $q_{T_2|X}(t_2 \mid x) = 1$. Notice that, for $x \in S$, we have that $q_{T_2|X}(\sigma(t_1) \mid x)$ is zero if $\sigma(t_1) \neq t_2$ (and is one otherwise). Therefore, $p(t_2 \mid t_1) = 0$ if $\sigma(t_1) \neq t_2$, and in case $\sigma(t_1) = t_2$ we have $p(t_2 \mid t_1) = \frac{\sum_{x \in S} q_{T_2|X}(\sigma(t_1)|x)p(x)}{\sum_{\dot{x} \in S} q_{T_2|X}(\sigma(t_1)|\dot{x})p(\dot{x})} = 1$. Hence $p(t_2 \mid t_1)$ can only take the values 0 and 1. From the expression for $H(T_2 \mid T_1)$ above, one concludes that $H(T_2 \mid T_1) = 0$. A similar argument holds for $H(T_1 \mid T_2)$. This shows that $\mathrm{VI}(T_1, T_2) = 0$. $\qquad\square$

## E   THE LEARNING ALGORITHMS

A natural way for us to construct a method for minimizing Equation (3) would be to replicate the procedure in Tishby et al. (2000) using an implicit analytical solution for $\nabla_{q_{T|X}} \mathcal{L}_{\mathrm{IB}}^{\beta} = 0$ to formulate the minimization of $\mathcal{L}_{\mathrm{IB}}^{\beta}$ as a multiple-minimization problem solvable by a coordinate descent algorithm (see Section 2.3). However, the expression resulting from $\nabla_{q_{T|X}} \mathcal{L}_{\mathrm{CIB}}^{\beta} = 0$ is much more complicated than that for $\nabla_{q_{T|X}} \mathcal{L}_{\mathrm{IB}}^{\beta} = 0$. Furthermore, the derivation in Tishby et al. (2000) relied on the fact that mutual information can be written as a KL divergence, but the causal information gain cannot (Simoes et al., 2024). So, although we did not show it is impossible to find a coordinate descent algorithm of the style of the Blahut-Arimoto algorithm, it is clear that a derivation of such an algorithm would have to take a very different form from the one in Tishby et al. (2000). Instead, we opted to find the minima of $\mathcal{L}_{\mathrm{CIB}}^{\beta}$ while staying constrained to the probability simplices using two types of projected gradient descent algorithms, which we now discuss.

Since $T$ is discrete, each conditional distribution $q_{T|X=x}$ can be seen as a vector of probabilities $(q_{T=t|X=x})_{t \in R_T}$ in the Euclidean space $\mathbb{R}^{|R_T|}$. Furthermore, since $X$ is also discrete, the encoder $q_{T|X}$ can be seen as a vector of probabilities

$$
(q_{T=t_1|X=x_1}, \ldots, q_{T=t_1|X=x_{|R_X|}}, q_{T=t_2|X=x_1}, \ldots, q_{T=t_{|R_T|}|X=x_{|R_X|}}) \in \mathbb{R}^{|R_T| \cdot |R_X|}.
$$

The $|R_T|$-dimensional subspace of $\mathbb{R}^{|R_T| \cdot |R_X|}$ corresponding to $q_{T|X=x}$ will be denoted $\mathbb{R}^{(\cdot|x)}$. Each conditional distribution vector $(q_{T=t|X=x})_{t \in R_T}$ must lie in the probability simplex $\Delta^{|R_T|-1}$, so that

$q_{T|X}$ must belong to the cartesian product of simplices $\Delta := \bigtimes_{x \in R_X} \Delta^{|R_T|-1}$. The causal information bottleneck problem can then be formulated as finding the global minimum of $\mathcal{L}_{\text{CIB}}^\beta[q_{T|X}]$ with $q_{T|X}$ constrained to $\Delta$. We will use projected Gradient Descent (pGD) and projected simulated annealing gradient descent (pSAGD). The pSAGD algorithm takes its inspiration from simulated annealing methods which help hill-climbing algorithms avoid local minima in discrete search spaces (Russell & Norvig, 2021). The pSAGD is a simulated annealing version of pGD which does something analogous. Being itself a local search algorithm, pSAGD still may converge to local minima. Re-running pSAGD a few times (effectively using an ensemble of pSAGD learners) will increase the likelihood of finding the global minimum.

**Projected Gradient Descent**   In projected gradient descent (pGD) (Bertsekas, 2016), each iteration step is of the form:

$$q_{T|X}^{(t+1)} = \Pi_\Delta \left( q_{T|X}^{(t)} - \alpha \nabla_{q_{T|X}} (\mathcal{L}_{\text{CIB}}^\beta) \Big|_{q_{T|X}^{(t)}} \right), \tag{19}$$

where $\Pi_\Delta$ is the Euclidean projection onto the constraint space $\Delta$ and $\alpha$ is the learning rate for the gradient descent step. It is easy to check that projecting onto $\Delta$ is equivalent to projecting each $q_{T|X=x}$ onto the probability simplex $\Delta^{|R_T|-1}$. In other words, we can apply the projection to each conditional distribution $q_{T|X=x} \in \mathbb{R}^{|R_T|}$ separately. That is,

$$q_{T|X=x}^{(t+1)} = \Pi_{\Delta^{|R_T|-1}} \left( q_{T|X=x}^{(t)} - \alpha \nabla_{q_{T|X=x}} (\mathcal{L}_{\text{CIB}}^\beta \big|_{\mathbb{R}(\cdot|x)}) \Big|_{q_{T|X=x}^{(t)}} \right). \tag{20}$$

This is illustrated in Figure 4. The Euclidean projection onto each probability simplex is done using our implementation of the algorithm for projection onto the probability simplex described in Duchi et al. (2008). Our implementation is vectorized, so that we can simultaneously apply the projection to each $q_{T|X=x}$ separately.

**Projected Simulated Annealing Gradient Descent**   In complex systems, multiple local minima of the CIB may exist. Projected simulated annealing gradient descent (pSAGD) addresses this issue by introducing randomness, or "jittering", into the gradient descent process, controlled by a temperature parameter. At each iteration $t$, instead of taking a step in the direction of the negative gradient (followed by a projection) at the current point $q^{(t)}$, there is a chance of jumping to a random neighbor, that is, a point $\tilde{q} \in \Delta$ obtained by uniformly sampled from the sphere centered on $q^{(t)}$ of radius equal to the learning rate (and applying a projection onto the simplex if necessary). The probability of accepting the proposed $\tilde{q}$ depends on the temperature and the quality of the proposal. Specifically, if the loss $L = \mathcal{L}_{\text{CIB}}^\beta$ at $\tilde{q}$ is lower than $L(q^{(t)})$, then $\tilde{q}$ is accepted, *i.e.*, $q^{(t+1)} = \tilde{q}$. Otherwise, the acceptance probability is given by $\exp(-\frac{L(\tilde{q}) - L(q^{(t)})}{T^{(t)}})$, where $T^{(t)}$ is the current temperature. The temperature decreases according to a chosen cooling rate $c \in (0, 1)$, such that $T^{(t+1)} = c \cdot T^{(t)}$. If $\tilde{q}$ is not accepted, a pGD step is taken instead.

**Implementation details**   We employed gradient clipping to prevent gradient explosions, which can destabilize the optimization process. Additionally, a cycle detection mechanism was incorporated, so that the optimization process stops if the algorithm cycles between two encoders, at which point the best solution is selected. We observed that using relatively large learning rates was necessary for effective training. This appears to stem from the small gradients encountered during optimization. Unsurprisingly, we also observed that, the higher the dimension of the search space, the larger the learning rates need to be. A cooling rate of $0.99$ was selected for the simulated annealing schedule, as it provides a reasonable decay curve for the probability of acceptance across different temperatures and typical step sizes, balancing the exploration and exploitation phases effectively. During the experiments, we noted that many local minima at which the optimizers got stuck corresponded to encoders that did not fully utilize the entire range of $T$. We call such representations non-surjective. We want to avoid such representations, since we would like the chosen range $R_T$ to be respected. To address this, we introduced a penalty term in the loss function that discouraged the optimizer from approaching non-surjective encoders. We tested this for $\gamma = 1$, where this adjustment greatly improved the accuracy of the optimizer, that is, the frequency at which it identified the ground truth.

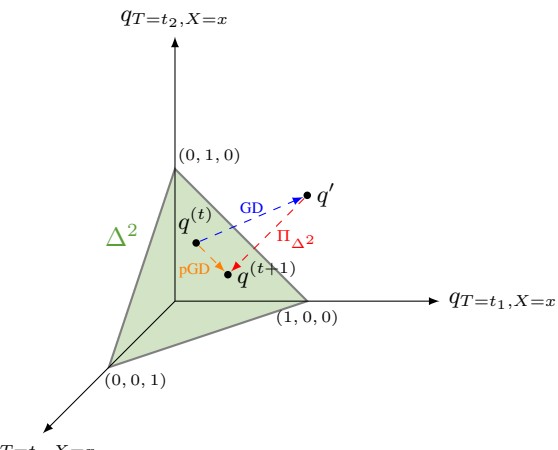

Figure 4: Illustration of a Projected Gradient Descent step in the $|R_T|$-dimensional slice $\mathbb{R}^{(\cdot|x)} \cong \mathbb{R}^{|R_T|}$ of $\mathbb{R}^{|R_T| \cdot |R_X|}$ corresponding to the conditional distribution $q_{T|X=x}$, for the case where $|R_T| = 3$. Here, $q^{(t)}$ denotes $q_{T|X=x}^{(t)}$ and $q'$ denotes the output of the Gradient Descent (GD) step, that is, $q' = q_{T|X=x}^{(t)} - \alpha \nabla_{q_{T|X=x}} (\mathcal{L}_{\mathrm{CIB}}^{\beta}\big|_{\mathbb{R}^{(\cdot|x)}})\big|_{q_{T|X=x}^{(t)}}$.

Despite achieving these high accuracies, we employed ensembles of optimizers to further enhance the likelihood of finding the global minimum. All experiments, along with the exploration of different parameter settings and additional results, are available in the accompanying code repository.

