# OpenReview forum: "Optimal Causal Representations and the Causal Information Bottleneck"
_ICLR.cc/2025/Conference — Submitted to ICLR 2025_

### Official Review · Reviewer_owLt · 2024-10-31

**Soundness:** 3
**Presentation:** 2
**Contribution:** 2
**Rating:** 6
**Confidence:** 3

**Summary:**

The authors propose an approach of learning causal representation from a ground truth distributions. In particular, the authors consider a setting in which causal relations are present, such that compression approaches that are unaware of relations might fail due to confounding effects. To overcome such confounding effects, the authors propose a causal information bottleneck (CIB) that leverages a causal information gain, opposed to standard information gain used in the normal IB, which considers changes in interventional distributions of the variables when intervening on the learned representation instead of standard conditioning. The authors propose intervention encoders and decoders, $p(T|X), q(T|X)$, that are learned to map between given variables $X$ and a representation $T$ while being regularized via the interventional distributions of the model. The resulting CIB guides optimization of the en-/decoders and therefore is able to balance the compression of variables, predictive power, and causal control.

The authors define a backdoor adjustment formula for the learned representation, transfers known identification results on standard SCM to the learned representation. The correct working of their approach is verified via three synthetic experiments, featuring different levels of confounding and representation compression. The results indicate a continuous transition between retained causal mutual information of the learned presentation and base model variables and the predictive performance of the model.

**Strengths:**

The paper is generally well written. Previous definitions, such as causal entropy and causal information gain, are well introduced and explained. The construction of the CIB resembles that of the standard IB and follows naturally. The Lagrangian formulation follows standard definitions and, while I'm not an expert on the topic, the applied optimizations seem be reasonable choices.

In comparison to prior works, the presented information-theoretic treatise provides a clear and defined trade-off between predictive performance, compression and retained causal information. The presented backdoor adjustment formula for representations is soundly defined and allows for identification of causal effects given that the adjustment set variables are known and can be conditioned on.

The authors successfully demonstrate a continuous transition between retained causal mutual information from the base model variables and the predictive performance of the model under a varying regularization factor.

**Weaknesses:**

The work generally overcomes several obstacles in learning causal abstractions via the proposed CIB. This, however, comes at the cost of strong assumptions of all discrete variables and having full access to their graph and joint distribution. While the authors mention the possibility to extend their work to continuous domain, experiments remain rather simple. Evaluations on the effects of incomplete or incorrect knowledge on the causal graph, as for example considered for the motivational example, or the extension to higher-dimensional domains, e.g. experiments on (synthetic) image data, are lacking.

To the best of my knowledge, the authors cite relevant works on causal abstraction and IB theory. However, prior work on applied causal representation learning is missing. This includes prior works on learning tau abstraction [1,2] and general identifiability results on latent causal representations, e.g. [3,4].


**Minor**

* The authors state that "the most common type of intervention is an atomic intervention". While this is certainly true within the field of causality (as it is the easiest to theoretically analyze), a commonly encountered problem in real-world settings is the inability to perform atomic interventions. The authors might want to tune this statement down a bit.
* Figure 1 is generally hard to read without zooming in. I would like to recommend increasing the text size and line widths if possible.


[1] Kekić, Armin, Bernhard Schölkopf, and Michel Besserve. "Targeted Reduction of Causal Models." *The 40th Conference on Uncertainty in Artificial Intelligence*. 2023.
[2] Massidda, Riccardo, Sara Magliacane, and Davide Bacciu. "Learning Causal Abstractions of Linear Structural Causal Models." *The 40th Conference on Uncertainty in Artificial Intelligence*.
[3] Zheng, Yujia, Ignavier Ng, and Kun Zhang. "On the identifiability of nonlinear ICA: Sparsity and beyond." *Advances in neural information processing systems* 35 (2022): 16411-16422.
[4] Wendong, Liang, et al. "Causal component analysis." *Advances in Neural Information Processing Systems* 36 (2024).

**Questions:**

I would like to ask the authors to address the weaknesses stated above. In addition, I got the following questions:

1) In the intervention decoder definition (eq. 4) the terms $p*$ and $q$ are defined as probabilities over over values $x, t$ and normalized over $x$. However, neither $p^*$ nor $q$ are conditioning on any particular intervention. While being named 'intervention decoder', it is unclear to me how interventions are exactly involved in the computation of these terms and I would like  to ask the authors to elaborate this.
2) The proposed backdoor adjustment formula marginalizes over variables $Z$ that satisfy the backdoor criterion. Given the motivational example that $Z$ might induce a selection bias and thus can not be fully observed, how realistic is the  scenario of the paper? Could the authors provide insights on the effects of missing or only partial observed confounders with regard to the performance of their approach?


**Rebuttal:** The authors presented an interesting approach, utilizing causal entropy and causal information gain to obtain a 'causally-aware' IB variant that might present a noteworthy addition to the overall tool box of causal training methods. While I find the presented approach interesting from a causal perspective, I also agree with the other reviewers on the rather limited scope of the conducted experiments in terms of practical applicability. The experiments seem to prove the correct working of the approach in small domains, but at the same time come with a number of assumptions, e.g. prior knowledge of the causal graph. Since all (C)IB methods rely on an auto-encoding property which might or might not be achieved in practice, I would have expected to see the actual application of the proposed method to a more complex setting. For the named reasons, after considering the reviews of the other reviewers and the authors' responses, I remain my score of a weak accept.

---

> ### Author Response · Authors · 2024-11-16
>
> Thank you for your review and questions.
>   We are happy to read that you found the paper generally well-written, and that it overcomes many of the obstacles in learning causality-aware abstractions.
> We will now address the weaknesses and the questions. There was a lot to address, so we will split our answer between comments.
>
> ## Weaknesses:
>
> > "The work generally overcomes several obstacles in learning causal abstractions via the proposed CIB. This, however, comes at the cost of strong assumptions of all discrete variables and having full access to their graph and joint distribution. While the authors mention the possibility to extend their work to continuous domain, experiments remain rather simple."
>
> 1. It seems that your main concern about our work is the strictness of our assumptions. The following comments explain why our assumptions are less restrictive than they may seem at first.
>
>     a. **Discrete variables assumption**: we made this choice since the main point of this paper was to propose a definition of _optimal causal representation_ (as a natural extension of optimal representation) and a new causal extension of the IB, introducing a sensible way to define representation interventions and test that it actually works. The complexity comes from the new concepts introduced, and if we used continuous variables this would introduce additional issues (e.g. relating to estimation of the (causal) information gain for continuous variables, which is known to be a challenging problem on its own right) that would, in our opinion, dilute the message, as they are irrelevant to the main contribution of the paper. That said, we agree that it would be important to extend this to continuous variables, and we see that as a natural extension of this work, for future papers. This mimics what happened with the standard IB, which was introduced for discrete variables and later extended to continuous ones (in different ways, and these extensions indeed introduced complexity that warranted their own papers).
>
>     b. **Assumption of knowledge of the causal graph**: indeed, one needs a causal graph to apply our method, either one hand-crafted by experts, or one constructed using a causal discovery algorithm.
>     We would like to point out that, without information of the causal relations/causal graph, no causal statements can be made.
>     In other words, we need information about our causal graph in order to build optimal causal representations.
>     So, although assuming that we have information about the causal graph is a strong assumption, we don't see it as much stronger than strictly necessary for our task - to build causality-aware abstractions we need to know causal relations.
>     One could indeed relax this assumption by assuming one knows only the parts of the graph that are necessary to render the conditional distributions necessary for computing the CIB identifiable.
>     If necessary, for a particular application with only partial graph knowledge, one could study whether the effect $p(y\mid \mathrm{do}(x))$ is identifiable.
>     However, identifiability issues are not the focus on this paper.
>     And, of course, one could come up with methods which learn causality-aware representations while simultaneously attempting to learn the graph, thus mixing causal discovery and causal representation learning tasks.
>     In general however, the causal discovery task is distinct from our task of learning optimal causal representations.
>     Our approach assumes the graph has already been learned.
>     Notice that, if we wanted to drop this assumption, we could always simply prepend the CIB method with our favorite causal discovery method.
>     Again, we thought that doing so would dilute the message.
>     We think that you raised a valid concern and, if accepted, we will explain clearly in the paper why we think this assumption is less restrictive than it may seem, and why to learn causality-aware representations you will always need to know (part of) the causal graph in some way, or that the user can prepend a causal discovery method of their choice to our method.
>
>     c. **Assumption of knowledge of the joint distribution**: similarly to the previous assumption, we can in principle prepend our method with a method which would learn the joint from data, but including this in our method would dilute the message. We would introduce the complexities of joint distribution estimation in our paper, unnecessarily complicating the exposition. Notice also that relaxing our assumption would force us to assume that we have enough data and a good method to estimate the joint, so that we would just be pushing the assumption somewhere else. Again, you have raised a valid concern and, if accepted, we will explain in the paper why we think this assumption is not as crippling an assumption as it may seem.
>
> (Continues in the next comment...)

---

> > ### Author Response · Authors · 2024-11-16
> >
> > > "Evaluations on the effects (...) higher-dimensional domains, e.g. experiments on (synthetic) image data, are lacking."
> >
> > 2. We agree that it would be interesting to see what happens if one has incomplete information or data, but we do not think of this as contributing to the main message we want to convey (which, as described above, consists of a definition of optimal causal representation, a minimization method to find it, and a definition of representation intervention that allows us to apply it), and see it as future work.
> >    And, certainly, higher-dimensional domains, such as in the case of image data, would also be interesting to study.
> >    That said, we chose smaller, easy to interpret examples where we can know the full-control ground truth, and which, in the case of the third experiment, can already encode non-trivial interactions between variables. This suffices to show that the method learns the representations we expect (thus providing evidence for the validity of our definitions and method), and illustrates how to apply it. Scaling this up, for example for image data, is not a focus of this paper, although we agree that it is an important direction for future research.
> >
> > > "To the best of my knowledge, the authors cite relevant (...) on latent causal representations, e.g. [3,4]."
> >
> > 3. Thank you for pointing out this extra literature. We can include these to make the Related Work section more complete. Please note, however, that their causal representation learning task is distinct from our task of learning optimal causal representations. That said, we see that this is important to mention in the paper explicitly as well.
> >
> > >"The authors state that 'the most common type of intervention is an atomic intervention'. While this is certainly (...) might want to tune this statement down a bit"
> >
> > 4. We agree. We will change this statement accordingly.
> >
> > >"Figure 1 is generally hard to read without zooming in. I would like to recommend increasing the text size and line widths if possible."
> >
> > 5. Yes, we can improve the visibility of the figure.
> >
> >
> > ## Questions:
> >
> > 1. Thank you for looking into such details. I assume that by "conditioning on a particular intervention" you mean a post-intervention distribution like $p(x|do(t))$. We could have chosen that notation, but, as just mentioned, usually this would be notation for a post-intervention distribution, which is not our intention here.
> >    Instead, what we mean is: given that the representation variable $T$ was set to $t$ by an intervention, then the probability that $X$ is set to $x$ is $p^*(x|t)$. This notation is very similar to that in Pearl's Causality, section 4.2:
> >     "We regard the stochastic intervention as a random process in which the unconditional intervention do(X = x) is enforced with probability $P^*(x | z)$. Thus, given $Z = z$, the intervention do(X = x) will occur with probability $P^*(x | z)$..."
> >    Also, notice that the definition of the intervention decoder $p^*(x|t)$ is compatible with the decoder $q$ in the sense that $q(t|x)=p^*(t|x)$; that is, the probability that the intervention $T$ was set to t given that $X$ was set to $x$ is $q(t|x)$.
> >    If accepted, we will add a sentence clarifying this.
> >
> > 2. We are not sure we understand the sentence "$Z$ might induce a selection bias and thus can not be fully observed": $Z$ may induce a bias and yet still be observed.
> >    That said, what we understand the question to be is: we used a backdoor criterion, which demands controlling for the variables in a backdoor set; in reality, how realistic is it that we can do this?". Some notes about this:
> >
> >     a. We first would like to point out that the CIB method does not need this to be the case: assumption 8 (which says that there is a backdoor set we can condition on) is purely to simplify the implementation of the algorithm. As we tried to make clear in the paper, if this assumption is broken, one can use do-calculus to find an expression for $p(y|\mathrm{do}(x))$ and thus for $p(y|\mathrm{do}(t))$. If even do-calculus fails (and we do not have access to the interventional data), then the method will indeed fail. But this is an issue with any causal method: if there is no way to compute the post-intervention distributions from the joint/observational distribution, there is nothing we can do.
> >
> >     b. To answer the question of how realistic is this scenario: indeed it often happens in practice that there are confounders which cannot be controlled for, and where there is no way (also using do-calculus as described above) to compute $p(y|\mathrm{do}(x))$ from the available data. This indeed makes it impossible to compute $I_c(Y|\mathrm{do}(T))$, and thus to use the CIB method. But we would like to reiterate: this is not exactly an issue with the CIB, but a known, fundamental issue in causality. If we do not have enough (causal) information about our system, we simply cannot make causal claims.

---

> > > ### Comment · Reviewer_owLt · 2024-11-25
> > >
> > > Dear Authors, thank you for your response. Especially, clarifying the overall role of identifiability and the difference between post-intervention probabilities and $p^*(x|t)$ helped my understanding.
> > >
> > > Regarding question 1: My question was (not solely) concerning the general strictness of assumptions, but the overall steps of the optimization procedure (e.g. how individual terms are computed). While the necessary formulas seem to be contained within the paper, I could imagine that clarity might be improved by providing the explicit steps of the method in form of a pseudo-code algorithm for the camera-ready version.
> > >
> > > Thank you once again for your answers. I still believe that the paper contains a valid contribution. However, given that the overall scope of experiments remains rather simple, I would like to remain with my current score.

---

> > > > ### Author Response · Authors · 2024-11-26
> > > >
> > > > Thank you for your thoughtful feedback.
> > > >
> > > > Indeed, although the necessary formulas are already included in the paper, we see that a concise, step-by-step pseudo-code could make the methodology even more accessible, and we will include this in the camera-ready version.
> > > >
> > > > Regarding the scope of the experiments: demonstrating that our method learns what it is designed to learn is possible when we have a ground-truth.
> > > > As we mentioned above, this is achieved by using relatively simple experiments, which can be easily understood.
> > > > An experiment of a much higher complexity than the Genetic Mutations experiment would not be amenable to easy interpretation of the results.
> > > > As you may be aware, employing simple experiments to illustrate a novel approach is not uncommon, particularly in research on causality.
> > > > For example, in [1] they exemplify their novel intervention selection method in very simple causal models, to transparently demonstrate their method’s validity and compare results with known ground-truths.
> > > >
> > > > We appreciate that you acknowledge the validity of our contribution, and we look forward to implementing these improvements for even greater clarity.
> > > >
> > > > [1] 2018 Lee, Sanghack and Bareinboim, Elias Structural causal bandits: Where to intervene?, NeurIPS

---

### Official Review · Reviewer_AP3s · 2024-10-31

**Soundness:** 2
**Presentation:** 2
**Contribution:** 3
**Rating:** 5
**Confidence:** 4

**Summary:**

The work extends the concept of information bottleneck to causal settings. It builds upon the idea of Tishby et al. and propose a new method for finding representations that respect the underlying causal mechanisms to a certain degree. A new definition of optimal causal representation is also presented. The work improves the interpretability of causal relationships and also extends the  interventional reasoning to complex causal systems.

**Strengths:**

1. Interesting work. Extending IB to a causal setting in a principled fashion is a novel task in my knowledge. The authors should be commended for this.

2. Impressive results (although in simple domains), but a good start nonetheless.

**Weaknesses:**

1. The paper talks about the representations being interpretable but the experimental evaluations do not justify the claims. Can the authors comment on this?

2. How are correct causal relationships obtained? I mean for simpler domains this is feasible but for more complex domains this can be a big issue.

3. Should'nt $\beta$  i.e. the Lagrange multiplier be learned rather than it being fixed?

4. Also for the interventional decoder, using a uniform prior seems to be a pretty generic choice. It would have been more interesting to see more complex priors.

**Questions:**

See the Weaknesses section.

---

> ### Author Response · Authors · 2024-11-16
>
> Thank you for your review and questions.
>   We are glad to read that you found our paper original, and the results impressive.
> We will now address the weaknesses and the questions:
>
> ## Weaknesses/Questions:
> 1. Because we use CIB instead of IB, the representation we construct is not only statistically associated, but it keeps causal control over $Y$.
> After a representation $T$ is learned for the full-control (i.e. $\gamma=1.0$ case), we may use it (and its encoder) to understand what parts of $X$ are more relevant than others to control $Y$. It is in this sense that $T$ can help to understand the relationship between $X$ and $Y$.
> Consider for example the last experiment (the Genetic Mutations experiment).
> As described in the paper, the encoder $q_{T \mid X}(t\mid x_1, x_2, x_3, x_4) = \delta_{t, \theta(x_1, x_2, x_3, x_4)}$ for the representation learned in the $\gamma=1.0$ case captures exactly the part of $X=(X_1,X_2,X_3,X_4)$ which influences $Y$.
> In particular, one can check that $q_{T\mid X}$ does not actually depend on $X_4$, which allows us to conclude that the $X_4$ component of $X$ has no control over $Y$, even though it is statistically associated with it. This is precisely the kind of thing that a non-causal method such as the IB would miss.
> One can also check, by analyzing $q_{T\mid X}$, that mutations on $X_3$ only have an effect on $Y$ if both $X_1$ and $X_2$ are mutated.
> Again, since $X_3$ and $Y$ are confounded through $S$, this complex interaction would not be captured by a non-causal method.
> Finally, one could also notice that $q_{T\mid X}$ is symmetric on $X_2$ and $X_1$, meaning that mutating either will result on the same effect on $Y$.
> All of these are conclusions about the causal relationship between $X$ and $Y$ made possible by the learned $T$.
> 2. This is indeed a big issue, usually called the causal discovery problem/task. This is however not the task that we tackle here. The causal discovery task is arguably the most famous problem of causality, but there are of course others: intervention selection, identification, causal abstraction learning, and so on. In contrast with the causal discovery task, other tasks often assume that the graph is given. In our task of learning _optimal causal representations_, we also assume the graph is given, either by experts, or another algorithm. If it is not given, then we really are in another domain of enquiry. We mentioned that the graph is given on line 092, but we can make all this clearer in the paper. That said, it would be interesting to study how to learn optimal causal representations when a graph is only partially known, but that would be a next step.
> 3. Beta is a tradeoff parameter that needs to be fixed by us. Beta controls a tradeoff between compression and (causal) sufficiency. How much compression we demand vs how much sufficiency we want to keep needs to be chosen by us - there is no principled way to do this. This is exactly what also happens with the standard IB.
> 4. The choice of uniform prior over interventions on $X$ corresponds to not knowing what interventions will be done on $X$ before choosing an intervention on $T$. This seemed like a natural modeling choice to us. If one had a specific particular case in mind where there was some prior knowledge about the intervener's "natural" choices, our method could accommodate that through a different choice of prior, but in our experiments we did not see a reason to assume any information about the interventions on $X$.

---

> > ### Comment · Reviewer_AP3s · 2024-11-24
> > **Response to the rebuttal**
> >
> > I would like to thank the authors for their response. I think there are a few important concerns that remain. For example,m in CRL neither the causal variabnles, nor the causal graph is known, so this assumption of known causal graph is a pretty strong one and one which can have a great impact on the overall problem setup. Also, the prior question remians to an extent. Even if your current set of experiments do not require it, will using another prior distribution be straightforward?
> >
> > Overall, I retain my score.

---

> > > ### Author Response · Authors · 2024-11-26
> > > **Our assumptions are as weak as possible for our specific task & The prior is straightforward to change**
> > >
> > > Thank you for continuing the discussion.
> > >
> > > About the prior: indeed implementing a different prior would be extremely straightforward: you would simply need to use the desired prior in the expression for $p(y\mid \mathrm{do}(t))$ in the algorithm.
> > >
> > > About the causal graph assumption: please note that *our task is a specific causal representation learning task*. As previously mentioned, we want to learn abstracted representations of $X$ which keep causal control over a target $Y$ (and thus extract the parts of $X$ which are causally relevant with respect to $Y$).
> > > This **requires** that the causal effect of $X$ on $Y$ is identifiable, which we guarantee with our assumptions. As we wrote to reviewer cUM2, our assumptions are basically as weak as they can be for our task. Comparing them to the assumptions used when learning causal representations from data is not a meaningful comparison.

---

### Official Review · Reviewer_cUM2 · 2024-11-02

**Soundness:** 3
**Presentation:** 3
**Contribution:** 2
**Rating:** 5
**Confidence:** 4

**Summary:**

Causal representation learning and causal abstractions are a key problem in causality and the paper proposes the Causal Information Bottleneck, a causal extension of the classical information bottleneck in representation learning.

The paper is nice to read and well structured. However both the theoretical contributions as well as the empirical contributions are very limited and I can thus not recommend the publication for acceptance in its current form.

**Strengths:**

Despite the rather negative review, I think it is a great topic and the paper addresses a good question. Causal abstractions are a key focus point and question for many problems. It is thus a very timely and very important problem. I really hope the authors continue with the direction of their work, and I enjoyed reading it. The paper is easy to follow, partially because the proofs are rather simple, and the experimental section is very short.

**Weaknesses:**

Wrt to baselines and empirical evaluation the authors write correctly that there are many more available e.g. ". Specifically, they use variational autoencoders (VAEs), relying on the fact that β-VAEs (when used for the task of reconstruction) minimize the information bottleneck (Achille & Soatto, 2018). However, their method does not account for causality, being limited to scenarios without confounding due to its reliance on the standard IB"
However the paper does not compare against VAEs, beta-VAEs, Graph-VAEs or even some approaches like [1,2] or any of the plenty variants which can be found by a quick google search.

[1] Subramanian, Jithendaraa, et al. "Learning latent structural causal models." arXiv preprint arXiv:2210.13583 (2022).
[2]Liu, Yuhang, et al. "Identifying weight-variant latent causal models." arXiv preprint arXiv:2208.14153 (2022).
[3] etc
While some of the above approaches might not be one to one applicable, it simply would need to be explained much more clearly why this specific setting is the first of its kind and not comparison is needed. While I agree that beta-vae is not a causal method on its own, I would still like to see it evaluated as well as other methods and see the benefit of the causal method on the task. In particular, I would not only want to see the causal graph recovered but used for a downstream task and then the comparison to beta vae or other methods applied to the downstream task.

What are the assumptions allowing causal discovery in this setting? Is faithfulness assumed? There are no statements about many of the established assumptions enabling causal discovery in the text, yet it is clear that causality can not be inferred from obervational data without further assumptions. How does the approaches assumptions related to the assumptions of classical approaches, and which ones are needed? Assumption 8 is introduced as the only assumption but that seems to simplify the problem a lot ...

Finally, if it is really just a toy case and the own method should be in the focus then clearly demonstrate the robustness to noise or some ablations.
Moreover, the provided propositions are really not propositions but follow directly from the definition, e.g. Proposition 6.

**Questions:**

See weakneses above.

---

> ### Author Response · Authors · 2024-11-16
>
> Thank you for your review and questions.
>   We are glad that you enjoyed reading our papers.
> We will address the weaknesses and the questions, in the order they were posed. There was a lot to answer to, so we will split it between comments.
>
> ## Weaknesses/Questions:
> > Wrt to baselines and empirical evaluation the authors write correctly that (...) to the downstream task.
>
> 1. Thank you for noticing that, the way that we wrote it, the sentence you quoted from the Related Work section may suggest that VAEs, $\beta$-VAEs, and other such methods are existing alternatives to ours. If that were the case, an experimental comparison would indeed be appropriate. However, this is not the case. VAEs and $\beta$-VAEs are representation learning methods whose goal is to construct a latent random variable $Z$ such that the observed data $X$ may have been generated from $Z$ by some generative process encoded in a (unknown) conditional distribution $p_{\theta}(x|z)$ (and $\theta$ needs to be learned). In contrast, our method aims to construct a representation $T$ of a variable $X$ which keeps causal control _over a chosen target_ $Y$ while compressing $X$. Notice that these are fundamentally different tasks, and there would be no clear way to apply VAEs or $\beta$-VAEs to our task. Furthermore, graph-VAEs (from [1]) is a method for learning the graph of latent variables (from which the data is assumed to be generated). Our method is not a graph-learning method; in particular, the goals of the CIB and the graph-VAE algorithms is different. Similarly, [2] proposes a method for learning SCM (graph + (linear, Gaussian additive noise) structural assignments) for the latent variables generating the data. This is not the task we are concerned with: we are not concerned with learning causal graphs nor structural assignments, and there is no assumption of existence of latent variables from which the data is generated.
>     Inspired by your comments, we realize that we need to make this distinction clearly in the paper: none of the existing algorithms solves the same task, so that an experimental comparison would not be informative, which is why we did not do it. Our task is to learn _optimal causal representations_ (which we define), in a natural extension to the task of learning optimal representations from the IB literature. If accepted, we will add this explanation to the paper.
>     The only existing (although non-causal) method that can be compared with the CIB is the IB method.
>     Therefore, again inspired by your comments, we are now working on re-running the experiments, but now minimizing the standard information bottleneck lagrangian, so that one can compare the IB with the CIB, to explicitly show how the advantage of our model over non-causal models, by showing that the CIB captures causality where the IB fails. We will hopefully get back to you during the rebuttal period when the results are in.
>     It is also noteworthy that the IB is a generalization of ($\beta$)-VAE when the target is $Y\ne X$ (see [3]), and this is also what [4] alludes to. Hence, by testing the IB, we are also testing the extension of the ($\beta$)-VAE in our representation learning task.
>
> > What are the assumptions allowing causal discovery in this setting? Is faithfulness assumed? There are no statements about many of the established assumptions enabling causal discovery in the text, yet it is clear that causality can not be inferred from obervational data without further assumptions. How does the approaches assumptions related to the assumptions of classical approaches, and which ones are needed?
>
> 2. There are indeed works that mix causal discovery with representation learning, but this is not the case in our work. There is no causal discovery assumptions needed, since we are not doing causal discovery. Please note that we are doing representation learning (in particular, we are learning a representation which is causally optimal in the sense discussed in the paper, and which extends the IB framework), but this is, as described in the previous point, a distinct task from the task in many causal representation learning papers (including [2] and [5] that you cited). If accepted, we will make this point clearer in the paper.
>
> (Continues in the next comment...)

---

> > ### Author Response · Authors · 2024-11-16
> >
> > > Assumption 8 is introduced as the only assumption but that seems to simplify the problem a lot ...
> >
> > 3. Assumption 8 (the existence of a backdoor set) is an identifiability assumption which is not necessary for the results of this paper, as we attempted to clarify in the last paragraph of Section 6. Rather, it helps when running the experiments.
> > Assumption 8 could be replaced by any other assumption that allows identification of the causal effect. Without any such assumption, it would be impossible to guarantee that a proposed representation has an identifiable causal control over $Y$. To avoid unnecessary complications relating to identifiability and do-calculus in the experiments (whose point is to check that reasonable representations can be learned using the CIB), we picked the backdoor assumption, because it is well-known and already quite versatile. If accepted, we will further clarify this in the paper.
> >
> > >  Finally, if it is really just a toy case and the own method should be in the focus then clearly demonstrate the robustness to noise or some ablations.
> >
> > 4. If we run experiments using the IB instead of the CIB, this will in fact simulate ablations: it will be equivalent to running CIB in cases where the confounders are latent. Furthermore, we agree that testing other values for the noise variables would enrich the paper, and we will work on this.
> >
> > > Moreover, the provided propositions are really not propositions but follow directly from the definition, e.g. Proposition 6.
> >
> > 5. Proposition 6 is named as such because of its relevance (and it is also not named a Theorem, which we agree may require a more substantial contribution). We see the simplicity of its proof as a feature, not a bug, and it illustrates that as long as we have an identification result about a graph, we can easily derive an expression for the post-representation-intervention distribution in terms of observational distributions.
> >
> > 6. _General Observation_: Please note that our main contribution is foundational. Namely, we present a novel definition of _optimal causal representation_ and a method to find it by extending the IB using a causal version of mutual information. We also propose a natural way to define representation interventions.  Existing approaches for generic task of learning representations of $X$ keeping relevant (causal) information about $Y$ ([6, 4, 7]) do not result in representations containing only the parts of $X$ that are causally/interventionally relevant with respect to $Y$, as described in the Related Work section, while our method does. We included experiments to illustrate how the method can be used, and to show that the method indeed finds what we expected, in simple cases where we have a ground truth.
> >
> > [1] 2018 He, Jiawei and Gong, Yu and Marino, Joseph and Mori, Greg and Lehrmann, Andreas; Variational autoencoders with jointly optimized latent dependency structure
> >
> > [2] Subramanian, Jithendaraa, et al.; "Learning latent structural causal models."
> >
> > [3] 2018 Achille, Alessandro and Soatto, Stefano Information dropout: Learning optimal representations through noisy computation
> >
> > [4] 2021 Holtgen, Benedikt; Encoding Causal Macrovariables
> >
> > [5] 2022 Liu, Yuhang and Zhang, Zhen and Gong, Dong and Gong, Mingming and Huang, Biwei and Hengel, Anton van den and Zhang, Kun and Shi, Javen Qinfeng; Identifying weight-variant latent causal models
> >
> > [6] 2000 Tishby, Naftali and Pereira, Fernando C and Bialek, William; The information bottleneck method
> >
> > [7] 2016 Chalupka, Krzysztof and Bischoff, Tobias and Perona, Pietro and Eberhardt, Frederick; Unsupervised discovery of El Nino using causal feature learning on microlevel climate data

---

> > > ### Comment · Reviewer_cUM2 · 2024-11-25
> > >
> > > Thanks a lot for your comments and especially for the provision of experiments with IB. I have read the reviews of others and some important concerns remain, but I have updated my score.

---

> > > > ### Author Response · Authors · 2024-11-26
> > > >
> > > > Thank you for taking the time to read our rebuttal and the discussion with the other reviewers as well.
> > > >
> > > > Would you be so kind as to enumerate what are your remaining concerns? Perhaps we can still address them. And if they all come from the comments of the other reviewers, we would like to know what points we have not, in your view, sufficiently addressed from those comments.

---

### Official Review · Reviewer_jhzR · 2024-11-02

**Soundness:** 4
**Presentation:** 4
**Contribution:** 3
**Rating:** 8
**Confidence:** 3

**Summary:**

The authors propose the Causal Information Bottleneck as an extension of the Information Bottleneck capable of respecting causal relationships between relevant variables. A formal context is provided before a (Lagrangian) minimization problem is proposed, using do-calculus. The manuscript concludes with experimental results on three examples of increasing complexity.

**Strengths:**

The manuscript is well written, with a good formal description of relevant concepts and illustrative computational examples. Additional details are expanded on in the appendix with more-than-adequate references provided. The concept of a Causal Information Bottleneck is introduced naturally as an "extension" of an Information Bottleneck, and it is shown that it can be leveraged, in an algorithmic way, to construct representations that respect causal representations between variables in a rigorous sense.

The material in the manuscript is original and, in my opinion, presented very clearly, with a good background in existing theory.

**Weaknesses:**

The examples provided are simple, which is helpful for understanding the relevant concepts; however it would be nice to demonstrate an application with a (considerably) larger set of variables or dependencies.
It is (somewhat) unclear how often the proposed solution approach will be expected to produce `good’ results in minimizing in minimizing equation (3).

Minor point: the term “representation” appears several times in the manuscript but is given a formal definition on pg. 6. The use of the term may be standard in the particular literature, but it would be constructive to at least reference the definition at first occurrence, or to use a different term whenever that definition is not applicable and “representation” is used more loosely.

**Questions:**

To what extent does one have to always check whether conditions (a), (b), and (c) are met each time one applies the proposed algorithm to a new problem? When would the proposed algorithm be expected to fail?

Can you comment on the scalability of the proposed algorithm?

Can you clarify the meaning of “conserve space” (line 62)?

---

> ### Author Response · Authors · 2024-11-16
>
> Thank you for your review and questions.
>   We are happy to read that you found our paper original and clearly presented.
> We will now address the weaknesses and the questions, in the order they appear:
>
> ## Weaknesses:
> > The examples provided are simple, which is helpful for understanding the relevant concepts; however it would be nice to demonstrate an application with a (considerably) larger set of variables or dependencies.
>
> 1. Please note that our main contribution is foundational. Concretely: we present a novel definition of _optimal causal representation_, a method to find it by extending the IB using a causal version of mutual information, and also propose a sensible way to define representation interventions. We included experiments to show that our method indeed finds what we expected. It finds the ground truth when it exists (gammas 0 and 1), and for other gammas finds results that are consistent between each other.
> The experiments and the results are included in the paper to show that our method is applicable and how it could be applied. We do not claim that our particular implementation (i.e. method for minimizing the CIB) is computationally efficient or scalable to applications with a large set of variables, though we do claim that the _CIB method itself_ can be applied to all applications regardless of the number of variables involved. So, to scale up the implementation is not a concern of this paper, although we agree it is a very natural question to ask, and an important direction for future research.
>
> > It is (somewhat) unclear how often the proposed solution approach will be expected to produce `good’ results in minimizing in minimizing equation (3).
>
> 2. The information about how often the proposed minimization method succeeds is indeed not included in the current version of the paper, although it is present in the supplementary material (in the IPython notebooks). We agree that this is relevant information, and will include this in the paper if accepted. Notice that this can only be meaningfully assessed for the full-control case (i.e. $\gamma=1.0$), where we have the ground-truth. For reference, the chosen minimization method finds the full-control ground truth in 100%, 98% and 95% of the runs for the Odd and Even, Confounded Addition and Genetic Mutations experiments, respectively.
> 3. About the minor point: thank for pointing this out to us. We agree and will make the suggested adjustment.
>
> ## Questions:
> 1. Note that (a), (b), and (c) are not conditions or restrictions on applicability. Rather, (a), (b), and (c) are the three things we want to verify about the experiment results of each experiment. As an answer to your question "When would the proposed algorithm be expected to fail?": the minimization algorithm that we chose to use fails when it gets stuck on a local minimum - a common issue for local search algorithms. In such cases, one may need to fine tune it, or use another minimization algorithm which may be more suitable for the case at hand.
> 2. We used a version of projected gradient descent (GD) with "jittering" (as described in the Appendix E). Being essentially a gradient descent algorithm, it can in principle be scaled up, but note that other alterations of GD may need to be used to avoid local minima in more complex experiments. Since this was not the focus on the paper, as said above, we did not look for other alternatives, since this sufficed to show that the CIB method finds what we expected.
> 3. By "conserving space", we mean that the representation is a compression of the variable it represents, in the sense of [1], for example. In practical terms, the representation uses less memory than the original variable.
>
>
> [1] 2000 Tishby, Naftali and Pereira, Fernando C and Bialek, William; The information bottleneck method

---

### Public Comment · ~Minyeol_Bae1 · 2024-11-15

I found the manuscript very interesting, particularly the proposed causal information bottleneck approach, which seems to present a promising new paradigm for causal inference. In order to gain a clearer understanding of the concepts, I have a couple of questions.

First, could you kindly explain the reasoning behind the definitions of causal entropy (Definition 12) and causal information gain (Definition 13)? I would appreciate it if you could elaborate on their operational meaning and significance.

Second, I would be grateful if you could clarify the relationship between causal information gain and the causal relationship between variables. How do these two concepts interrelate?

Thank you very much for your time and consideration.

---

> ### Author Response · Authors · 2024-11-21
>
> Thank you for your interest and the questions.
> Causal entropy $H_c(Y|\mathrm{do}(X))$ measures the entropy that $Y$ has, on average, if one intervenes on $X$. $I_c(Y|\mathrm{do}(X))$ is the decrease in entropy, on average, if one interves on $X$. The latter was introduced to measure causal control of $X$ over $Y$. Please note that these quantities were not introduced in this paper for the first time. More details and further discussion/motivation can be found in the appendix A.2 and the papers cited in that section.

---

### Author Response · Authors · 2024-11-22
**Experimental comparison with the Information Bottleneck**

Inspired by some of your comments, we decided to run the Information Bottleneck (IB) method on the three experiments, in order to compare the results with those that we had obtained using the CIB method, and confirm experimentally that the IB method fails in our task of learning optimal causal representations. We would like to emphasize (as we had said in some of the comments) that no other existing method (in particular those from the causal representation learning literature) would be suitable for a comparison with the CIB method, since our task is fundamentally different.
The results were as expected: the representations learned with the IB method do not successfully capture the aspects of $X$ which causally affect $Y$ whenever there is confounding. The details can be found below.

The representation learned using the IB method for the Confounded Addition experiment was the following map:

| X1 | X2 | T |
|----|----|---|
|  0 |  0 | 0 |
|  0 |  1 | 0 |
|  1 |  0 | 1 |
|  1 |  1 | 2 |

Notice that this erroneously (from a causal perspective) maps the cases $(X_1 = 0, X_2 = 1)$ and $(X_1 = 1, X_2 = 0)$ to different values of $T$, even though these two cases result in exactly the same causal effect on $Y$ (but not the same (non-causal) predictive power over $Y$, due to the confounding through $W$).

For the Genetic Mutations experiment, the representation learned using the IB method was the following map:
| X1 | X2 | X3 | X4 | T |
|----|----|----|----|---|
|  0 |  0 |  0 |  0 | 1 |
|  0 |  0 |  1 |  0 | 1 |
|  0 |  1 |  0 |  0 | 3 |
|  0 |  1 |  1 |  0 | 3 |
|  1 |  0 |  0 |  0 | 3 |
|  1 |  0 |  1 |  0 | 3 |
|  1 |  1 |  0 |  0 | 2 |
|  1 |  1 |  1 |  0 | 2 |
|  0 |  0 |  0 |  1 | 1 |
|  0 |  0 |  1 |  1 | 0 |
|  0 |  1 |  0 |  1 | 3 |
|  0 |  1 |  1 |  1 | 3 |
|  1 |  0 |  0 |  1 | 3 |
|  1 |  0 |  1 |  1 | 3 |
|  1 |  1 |  0 |  1 | 2 |
|  1 |  1 |  1 |  1 | 2 |


Notice that, in contrast with the ground truth $\underline{T}$ (which is successfully learned by the CIB), this representation fails to distinguish the cases $(X_1 = 1, X_2 = 1, X_3 = 0, X_4 = x_4)$ and $(X_1 = 1, X_2 = 1, X_3 = 1, X_4 = x_4)$; that is, it fails to capture the protective effect of $X_3$. Hence, the IB method fails to learn the complex, epistatic interactions between the mutations that the CIB successfully learns. Notice further that the representation learned by the IB is not invariant on $X_4$. This is due to the confounding between $X_4$ and $Y$, which the CIB, in contrast, successfully handles.

In the Odd and Even experiment there is no confounding. The effect of interventions on $X$ are indistinguishable from the effect of conditioning on $X$, so that  we expect the representation with the most causal control and the representation with the most predictive power over $Y$ to coincide. This is indeed what happens: the representation learned by the IB in the Odd and Even experiment is exactly the same as the one learned by the CIB, as expected.

---

### Author Response · Authors · 2024-12-04
**Final remarks**

We would like to thank all reviewers again for all their thoughtful remarks and discussion points.

To summarize, we would like to stress that our task falls under the general problem of "causal representation learning" (CRL), but is different from the standard task. As a result, we have no other methods to compare to in the experimental evaluation; rather, our experiments aim to demonstrate the validity of our method. Simple experiments lend themselves to intuitive interpretations and existence of ground-truth representations which allows us to do this. Furthermore, our assumptions are as weak as they can be for this task, and it is not meaningful to compare them with the assumptions of methods tackling other tasks (including methods tackling other CRL tasks).

---

### Meta-Review · Area_Chair_YrxF · 2024-12-21

**Metareview:**

The paper proposes the Causal Information Bottleneck (CIB) as an extension of the classical Information Bottleneck, incorporating causal relationships between variables using causal information gain and interventional distributions.

Strengths

+ Provides an extension of the Information Bottleneck to causal settings



Weaknesses

+ Lacks strong empirical validation on large, complex domains or real-world applications.

+ Lacks comparison with existing approaches like causal abstraction methods (e.g. causal feature learning (K Chalupka, F Eberhardt, P Perona - Behaviormetrika, 2017), which targets optimal coarsening of information) and other causal (or non-causal) representation learning methods.

+ Assumes access to complete causal graphs and discrete variables, which reduces the applicability to more general settings without additional robustness studies

**Additional Comments On Reviewer Discussion:**

The reviewers agree that the paper presents interesting ideas but the ideas are not fully explored yet and not put into adequate context.

---

### Decision · Program_Chairs · 2025-01-22

Reject